# VectorWorld:
# Efficient Streaming World Model via Diffusion Flow on Vector Graphs

**Chaokang Jiang**[1]  **Desen Zhou**[1]  **Jiuming Liu**[2]  **Li Sun**[1]

## Abstract

Closed-loop evaluation of autonomous-driving policies requires interactive simulation beyond log replay. Existing generative world models suffer three gaps: history-incompatible initialization, sampling latency exceeding real-time budgets, and compounding kinematic infeasibility. We propose VectorWorld, a streaming vector-graph world model that incrementally generates ego-centric lane–agent tiles during rollout. VectorWorld couples a motion-aware gated VAE for history-compatible initialization, an edge-gated relational DiT with interval-conditioned MeanFlow and JVP-based large-step supervision for solver-free outpainting, and $\Delta$Sim, a physics-aligned NPC policy with hybrid discrete–continuous actions and differentiable kinematic logit shaping. On Waymo Open Motion and nuPlan, VectorWorld improves map fidelity, initialization validity, and density calibration, enabling stable real-time $1\,\mathrm{km}+$ closed-loop rollouts.

## 1. Introduction

Closed-loop evaluation and training of autonomous-driving policies require interactive simulations that extend beyond recorded logs. Log replay can neither answer counterfactual queries nor extend the horizon once the recorded context ends (Caesar et al., 2021; Gulino et al., 2023). Although generative world models (Ha & Schmidhuber, 2018; Feng et al., 2023; Tan et al., 2023; Yang et al., 2025a) can synthesize diverse lane–agent scenes, deploying them as an *online* simulator backend introduces constraints (initialization, latency, and long-horizon stability) that standard open-loop metrics do not capture (Tan et al., 2025).

Project page: `jiangchaokang.github.io/VectorWorld/`. [1]Bosch, Shanghai, China [2]Department of Engineering, University of Cambridge, Cambridge, United Kingdom. Correspondence to: Li Sun <sen5sgh@bosch.com>.

*Proceedings of the 43rd International Conference on Machine Learning*, Seoul, South Korea. PMLR 306, 2026. Copyright 2026 by the author(s).

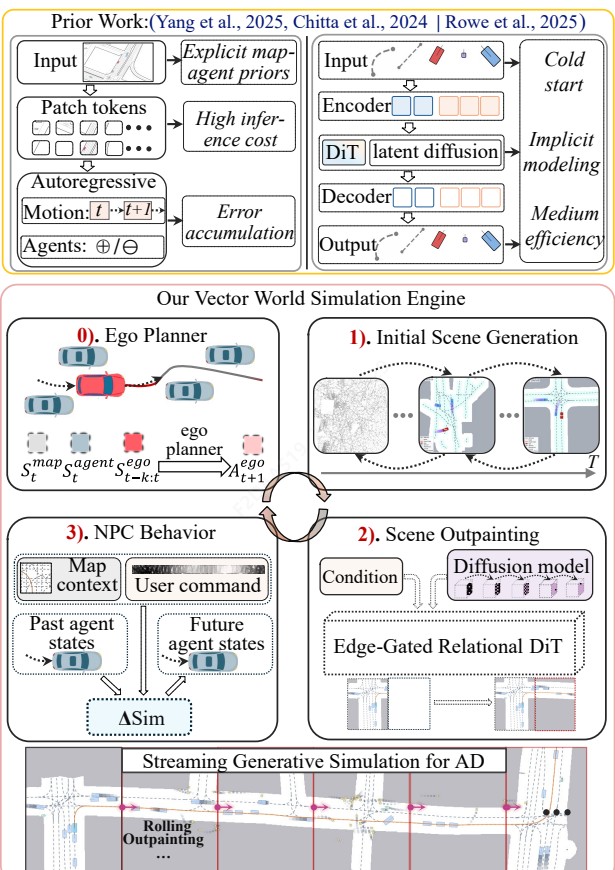

*Figure 1.* **Comparison with Prior Works.** Unlike prior pipelines that degrade in closed loop due to accumulated drift (autoregressive rollout) or history-free cold-start initialization, VectorWorld supports kilometer-scale closed-loop simulation via streaming outpainting in our experiments. The engine operates in a closed loop: 0) An Ego Planner drives within the scene; 1) Initial Scene Generation creates the base tile; 2) Scene Outpainting dynamically extends the map frontier using edge-gated relational DiT; and 3) NPC Behavior ($\Delta$Sim) ensures reactive interactions. This mechanism enables consistent, kilometer-scale driving simulation by progressively outpainting the world ahead.

We identify three constraints that dominate closed-loop simulation but are weakly reflected by standard open-loop metrics. First, history-free initialization: many generators output a static snapshot at $t = 0$ (Rowe et al., 2025), while

reactive policies are conditioned on short motion histories, inducing cold-start transients (e.g., jerk spikes) and unstable early interactions. Second, sampling latency: diffusion-based solvers require many denoising steps (Chitta et al., 2024; Tan et al., 2025) for structured lane–agent graph generation, which is incompatible with streaming outpainting under a real-time budget. Third, long-horizon feasibility: minor per-step kinematic violations, while negligible in short-horizon metrics, accumulate over kilometer-scale rollouts through compounding drift (Rowe et al., 2024; Lin et al., 2025).

To address these constraints, we propose VectorWorld (Figure 1), a streaming world model built on vector-graph representations. For initialization, we introduce an interaction-state interface that augments each agent with a compact motion-history code learned by a motion-aware gated VAE, enabling warm-start inputs for history-conditioned policies. For real-time outpainting, we propose an edge-gated relational DiT trained with interval-conditioned MeanFlow (Geng et al., 2025) and a JVP-based large-step objective, enabling solver-free (without iterative diffusion/ODE solvers) one-step latent sampling while preserving lane connectivity and agent–lane relations.

**Contributions.** We present VectorWorld as a real-time vectorized closed-loop simulator. Its lower-level mechanisms are not independent add-ons; they are the three interfaces required to satisfy the deployment constraints above:

- **History-compatible initialization.** A motion-aware gated VAE produces agent interaction states that warm-start history-conditioned ego and NPC policies, reducing initialization collisions $9.45 \to 4.69$ and jerk $16.6 \to 9.6$.

- **Real-time structured frontier outpainting.** Lane and agent tokens are jointly completed in masked vector-graph latent space using an edge-gated relational DiT trained with interval-conditioned MeanFlow and JVP-based large-step supervision, enabling solver-free one-step sampling at $\approx 5.6$ ms per single candidate tile; the 8-candidate closed-loop batch costs $\approx 44.8$ ms per outpainting trigger.

- **Long-horizon feasible interaction.** A one-pass NPC policy with hybrid discrete–continuous actions and differentiable kinematic alignment reduces compounding errors over kilometer-scale rollouts, improving ADE $2.80 \to 1.72$, lateral-acceleration violation $14.2\% \to 2.1\%$, and forward passes $2 \to 1$.

Experiments on Waymo Open Motion and nuPlan show that these modules jointly improve initialization validity, local map continuity, density calibration, topology fidelity, and $1\,\mathrm{km}+$ closed-loop policy evaluation and training.

**Conflict of Interest Disclosure.** C. Jiang, D. Zhou, and L. Sun are employed by Bosch; J. Liu is affiliated with the University of Cambridge. All experiments are conducted on publicly available academic benchmarks, namely Waymo Open Motion (Gulino et al., 2023) and nuPlan (Caesar et al., 2021). This work does not evaluate or promote any product, dataset, or service developed by Bosch or the University of Cambridge. The authors declare no financial conflicts of interest related to this work.

## 2. Related Work

**Driving World Models and Vectorized Scene Generation.** Recent driving world models generate future observations from history sensor inputs (e.g., videos) (Gao et al., 2024; Chen et al., 2025; Ni et al., 2025), while other approaches generate simulation environments from perception outputs, such as bounding boxes and lane graphs (Chitta et al., 2024; Sun et al., 2024; Jiang et al., 2024c; Tan et al., 2025). Several methods focus on initial scene generation (Rowe et al., 2025), generating agents, lanes, or both, sometimes via rasterization, which increases computational cost. Vectorized representations (Jiang et al., 2023; Chen et al., 2024) better match map geometry and can be more efficient for structured generation. Scenario-centric systems further combine initial scene generation with closed-loop behavior simulation, often decoupling the two for flexibility. Our approach complements these lines by introducing an *interaction-state* representation that reduces initialization mismatches, and by enabling *one-step* (or few-step) graph-latent generation suitable for online inpainting and outpainting. Figure 1 summarizes these deployment gaps (cold-start mismatch, solver latency, and long-horizon drift) and positions VectorWorld relative to representative prior pipelines.

**Fast Sampling in Diffusion and Flow-based Methods.** Denoising diffusion probabilistic models (DDPM) (Ho et al., 2020) and classifier-free guidance (Ho & Salimans, 2022) have become a standard approach for high-fidelity generation, and latent diffusion (Podell et al., 2023) reduces sampling costs by operating in a compressed space. Transformer backbones such as DiT (Peebles & Xie, 2023) improve scalability and conditioning flexibility. A large body of work accelerates sampling, including DDIM (Song et al., 2020), DPM-Solver (Lu et al., 2022), EDM (Karras et al., 2022), and one-step consistency-style methods (Song et al., 2023). Flow Matching (Lipman et al., 2022) and Rectified Flow (straight-line probability paths) (Lee et al., 2024) provide a training alternative that often admits fewer solver steps. MeanFlow (Geng et al., 2025) aims to learn interval-conditioned velocity parameterizations that enable solver-free one-step sampling. Our work differs in two ways: (i) we operate on *heterogeneous vector-graph* latents for maps and agents, and (ii) we unify DDPM/Flow/MeanFlow in

one edge-gated relational DiT backbone with dual-time conditioning, using a JVP-augmented MeanFlow objective to make one-step structured completion practical.

**Closed-Loop Behavior Simulation.** Closed-loop multi-agent simulation requires reactive behavior models that remain stable under rollout. Prior work spans learned agent simulators (Igl et al., 2022; Suo et al., 2023; Yang et al., 2025b), next-token traffic modeling (Philion et al., 2024), and return-conditioned or reactive policies such as CtRL-Sim (Rowe et al., 2024). While large Transformers and diffusion-based policies can improve realism, their autoregressive paradigm can incur high costs for long rollouts. Classical simulators (e.g., CARLA (Dosovitskiy et al., 2017), SUMO (Krajzewicz et al., 2012)) and procedural systems (e.g., MetaDrive (Li et al., 2021)) often rely on heuristics for agent insertion/removal, which can miss complex real-world interactions. Compared to them, VectorWorld focuses on stabilizing the full closed loop by (i) aligning the world-model output with policy inputs via interaction states, and (ii) enforcing feasibility via differentiable physics-guided logit shaping and inference-time constraints, with training–inference alignment to mitigate exposure bias.

## 3. Method

**Method Overview.** VectorWorld is organized as a streaming closed-loop pipeline with three deployment-facing interfaces. First, the interaction-state interface converts a generated agent snapshot into a policy-compatible state by pairing a compact static state with a body-frame motion-history code (Figure 2). Second, the edge-gated relational DiT completes masked lane and agent latents at the map frontier while preserving edge-defined topology and agent–lane alignment. Third, $\Delta$Sim rolls out NPCs with physics-aligned actions so that small kinematic errors do not compound over long horizons. This section defines the pipeline in runtime order: vector-graph latent representation, interaction-state initialization, relational latent generation with interval-conditioned MeanFlow, streaming outpainting, and NPC behavior simulation.

### 3.1. Vector-Graph Representation

We represent each ego-centric scene tile as a heterogeneous vector graph $\mathcal{G} = (\mathcal{V}, \mathcal{E})$ with lane nodes $\mathcal{V}_{\text{lane}}$ and agent nodes $\mathcal{V}_{\text{agent}}$. Edges are typed by interaction semantics: $\mathcal{E} = \mathcal{E}_{\text{L2L}} \cup \mathcal{E}_{\text{A2A}} \cup \mathcal{E}_{\text{L2A}} \cup \mathcal{E}_{\text{A2L}}$, capturing lane connectivity (predecessor/successor/adjacent) and lane–agent relations (proximity and alignment). Compared to rasterization (Chitta et al., 2024), vector graphs keep topology explicit and make missing regions well-posed: we can clamp observed tokens and complete only frontier tokens, which is the key operation in streaming outpainting.

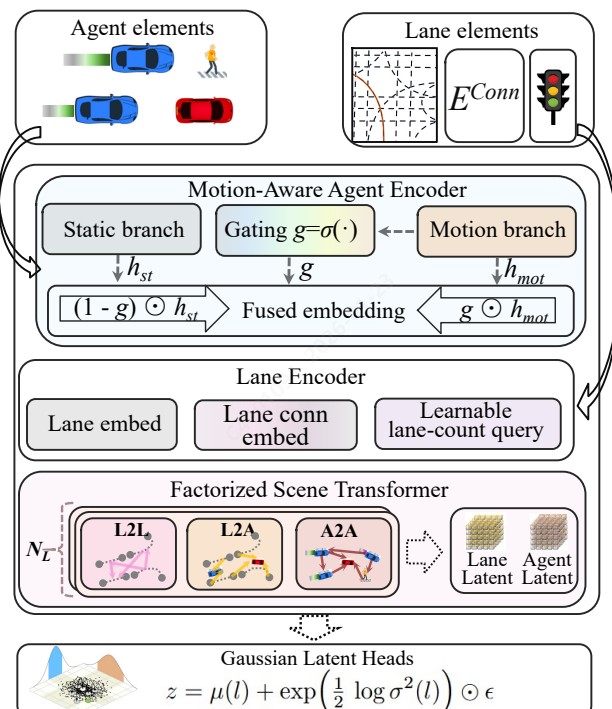

*Figure 2.* **Interaction-state interface with motion-aware gated VAE.** The encoder embeds lane elements, lane connectivity, and agent elements into Gaussian latents for downstream generation. For each agent, a motion-aware gate $g_i$ adaptively fuses a static branch from the instantaneous state $s_i$ and a motion branch from the body-frame motion code $m_i$, suppressing history noise for stationary agents while preserving short-horizon dynamics for moving agents. A factorized scene transformer aggregates typed relations (L2L / L2A / A2A), producing $z = (z_{\text{lane}}, z_{\text{agent}})$ as the latent interface used by initialization and streaming outpainting.

We learn a latent interface for $\mathcal{G}$ via an autoencoder. Let $z = (z_{\text{lane}}, z_{\text{agent}})$ denote the lane/agent latent tokens produced by the encoder. Generation, inpainting, and outpainting are all implemented as masked completion in this latent space, enabling a single generator to serve both initialization ($t = 0$) and rollout-time outpainting.

### 3.2. Interaction-State Interface

**Motivation: History-Conditioned Policies.** Closed-loop behavior policies, including many ego planners and NPC models, are history-conditioned because intent inference depends on recent velocity, curvature, and braking trends. Consequently, a history-free snapshot at initialization induces an implicit near-zero history prior for downstream policies, often resulting in large jerk and unstable early interactions.

**Interaction State: Static + Motion Code.** We therefore instantiate each agent $i$ through an interaction-state interface: a static state $s_i \in \mathbb{R}^7$ in the unified format

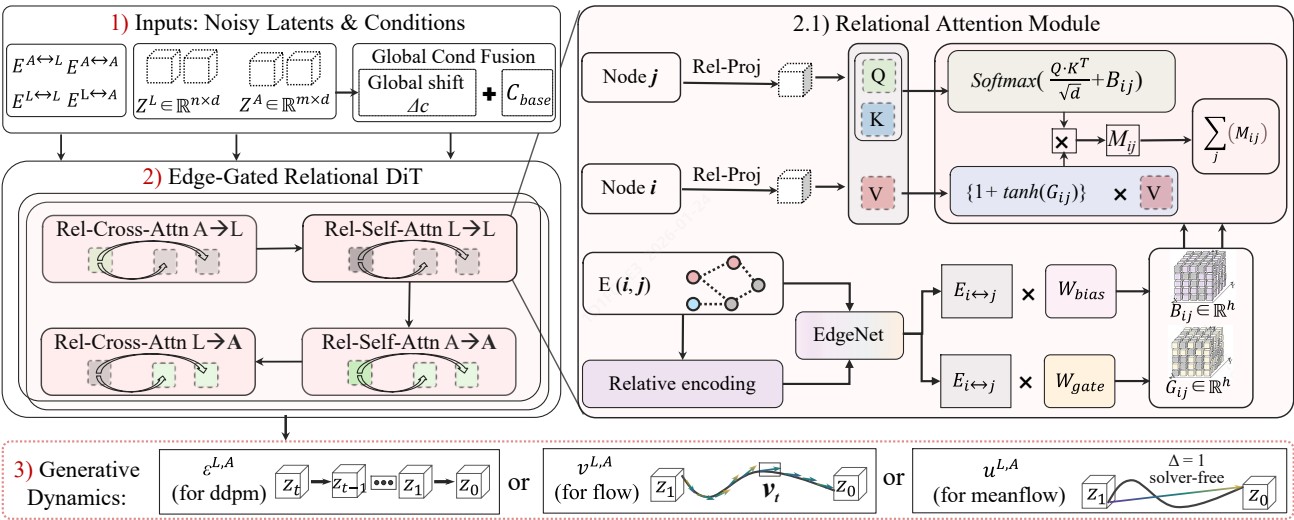

*Figure 3.* **Edge-gated relational DiT for vector-graph latent generation.** The framework generates vector-graph latents via a factorized transformer backbone. To capture complex graph dependencies, we introduce a Relational Attention Module (Right). Unlike standard attention, this module conditions message passing on edge features $\mathbf{e}_{ij}$ via: 1) An additive bias $B_{ij}$ applied to attention scores to regulate connectivity; 2) A multiplicative gate $G_{ij}$ applied to value vectors to modulate feature aggregation. This design enables precise structural control across heterogeneous agents and map elements, compatible with various generative dynamics (DDPM, Flow, or MeanFlow).

$[x, y, \text{speed}, \cos\theta, \sin\theta, \text{length}, \text{width}]$, a type token $\tau_i$, and a motion-history code $m_i \in \mathbb{R}^{2K_{\text{mot}}}$ represented as a $K_{\text{mot}}$-point polyline in the agent frame. At simulation start, $m_i$ is deterministically unrolled into a short pseudo-history sequence to fill the policy context window (Section A.4).

**Selective Motion Encoding.** Motion history exhibits sparse information content: for parked or slow-moving agents, aggressively reconstructing $m_i$ can inject unnecessary motion noise into the latent interface. We therefore use the motion-aware gated VAE shown in Figure 2. The learned gate activation empirically correlates with motion magnitude (Appendix Figure 7), producing an interpretable separation between static and dynamic agents:

$$g_i = \sigma\Big(f_{\text{gate}}\big([s_i, m_i, \tau_i]\big)\Big),$$
$$h_i = (1 - g_i) \odot f_{\text{st}}\big([s_i, \tau_i]\big) \ + \ g_i \odot f_{\text{mot}}\big([m_i, \tau_i]\big), \ (1)$$
$$q_\phi(z_i \mid s_i, m_i, \tau_i) = \mathcal{N}\Big(\mu_\phi(h_i), \ \text{diag}\big(\sigma_\phi(h_i)^2\big)\Big),$$

where $g_i \in (0, 1)^d$ is a learned gate, $\sigma(\cdot)$ is the sigmoid, and $\odot$ denotes element-wise product.

**Scene-Level Relational Encoding.** Following the factorized design detailed in Appendix A.5, lane and agent tokens are processed by a factorized scene transformer that alternates L2L, L2A, and A2A interactions, and uses a learnable lane-count query for masked completion. This design produces generation-friendly Gaussian latents $z$ whose agent component is selectively history-aware, enabling warm starts without introducing motion-code noise

for static agents. This VAE defines the latent interface $z$ that the outpainting generator completes in Section 3.3. We provide the full $\beta$-VAE objective in Appendix A.5.

### 3.3. Edge-Gated Relational DiT

**Edge-Consistent Completion.** Let $z = (z_{\text{lane}}, z_{\text{agent}})$ be VAE latents. We learn a generator that completes missing latent tokens given observed tokens and context $c$ (counts, and optional scene labels). The challenge is that topology and cross-type alignment are edge-defined: lane traversability depends on $\mathcal{E}_{\text{L2L}}$, while realistic interactions require agent–lane consistency. Node-only attention can match marginal token statistics yet violate these constraints, and such violations are amplified when masked completion is invoked repeatedly during streaming outpainting.

**Factorized Typed Attention.** To this end, we adopt a factorized DiT backbone (Figure 3) that alternates attention blocks over typed edge sets $\mathcal{E}_{\text{L2L}}, \mathcal{E}_{\text{A2A}}, \mathcal{E}_{\text{L2A}}, \mathcal{E}_{\text{A2L}}$. This explicitly matches heterogeneous interactions while keeping complexity compatible with real-time outpainting.

**Edge-Conditioned Bias and Value Gating.** For each typed edge $(i, j)$, we compute a relative edge embedding $\mathbf{e}_{ij}$ and modulate attention through an additive bias and a

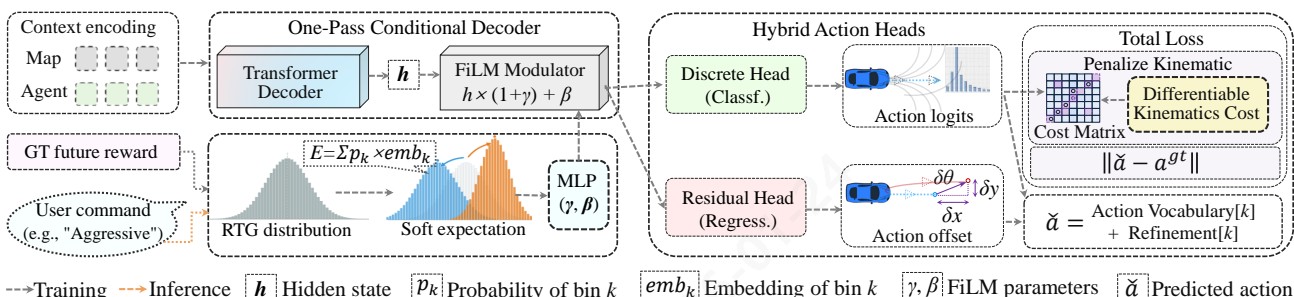

*Figure 4.* △**Sim: physics-aligned NPC policy.** A single-pass return-to-go (RTG) control embedding modulates the decoder via FiLM $(\gamma, \beta)$. Actions use a hybrid head (discrete k-disks token plus continuous residual) with differentiable kinematic logit shaping and DKAL regularization. This design reduces feasibility violations that compound under kilometer-scale closed-loop simulation.

multiplicative value gate:

$$
\alpha_{ij}^{(h)} = \mathrm{softmax}_j \left( \frac{\langle q_i^{(h)}, k_j^{(h)} \rangle}{\sqrt{d_h}} + B^{(h)}(\mathbf{e}_{ij}) \right),
$$
$$
y_i^{(h)} = \sum_j \alpha_{ij}^{(h)} \left( G^{(h)}(\mathbf{e}_{ij}) \odot v_j^{(h)} \right),
$$
(2)

where $h$ indexes attention heads, $d_h$ is the head dimension, $B^{(h)}(\cdot)$ is an edge-conditioned logit bias, and $G^{(h)}(\cdot)$ is an edge-conditioned gate (initialized as identity to preserve optimization stability).

**Mechanistic Effect.** The bias term shapes *connectivity preferences*, while the gate controls *message strength* by suppressing edge-inconsistent feature aggregation. This directly targets the main failure mode of streaming: cumulative structural drift under repeated masked completion, improving lane connectivity preservation and lane–agent alignment without rasterization or heuristic stitching.

### 3.4. Interval-Conditioned MeanFlow

**Constraint: Real-Time Large-Step Transport.** Streaming outpainting calls the generator during rollout and thus requires real-time inference. Multi-step diffusion or ODE solvers are too costly to invoke repeatedly. We therefore train the generator to support *large-step* transport, including the solver-free one-step case that we deploy online (Figure 5). We use $k$ for discrete simulation steps and $t \in [0, 1]$ for latent-generation time; $\Delta$ denotes the MeanFlow interval length (full notation in Appendix A.2).

**Rectified Path with Masked Clamping.** We use the straight-line probability path between data latents $z$ and Gaussian noise $\epsilon \sim \mathcal{N}(\mathbf{0}, \mathbf{I})$:

$$
z_t = (1 - t) z + t \epsilon, \qquad t \in [0, 1]. \quad (3)
$$

For masked (conditioned) tokens, we enforce a constant path by setting $\epsilon = z$, which implies a zero target velocity

and prevents conditioned structure from drifting during completion. Along this path, the rectified-flow target velocity for unmasked tokens is $v^\star = \epsilon - z$.

**Interval-Conditioned Mean Velocity and JVP Correction.** Standard rectified flow matching supervises an instantaneous velocity field, which is accurate under small solver steps but does not directly constrain single large-step transport. MeanFlow instead predicts an *interval-averaged* velocity $u_\theta(z_t; t, \Delta, c)$ over an interval of length $\Delta = t - r$ (with $r \leq t$), conditioned on scene context $c$ (counts and optional scene labels). The model is parameterized with two time channels $(t, \Delta)$. To improve accuracy at large steps (and in particular at the one-step boundary $(t, \Delta) = (1, 1)$), we supervise a first-order corrected predictor and minimize its mean-squared error against the rectified-path target:

$$
V_\theta(z_t; t, \Delta, c) = u_\theta(z_t; t, \Delta, c) + \Delta \left. \frac{\mathrm{d}}{\mathrm{d}t} \right|_\Delta u_\theta(z_t; t, \Delta, c),
$$
$$
\mathcal{L}_{\mathrm{mf}} = \mathbb{E} \left[ \| V_\theta(z_t; t, \Delta, c) - v^\star \|_2^2 \right],
$$
(4)

where the derivative is evaluated with the interval-length channel $\Delta$ held fixed: it differentiates through both the explicit time input and the implicit dependence on $z_t$, but not through the step-size conditioning input $\Delta$. We implement this derivative efficiently as a Jacobian–vector product (JVP); see Appendix A.8 for the exact training step.

**Guided Solver-Free Update.** For conditional generation, we apply classifier-free guidance (CFG) on the mean-velocity field:

$$
u_\theta^{\mathrm{cfg}} = u_\theta(z_t; t, \Delta, c_\varnothing)
$$
$$
+ s \left( u_\theta(z_t; t, \Delta, c) - u_\theta(z_t; t, \Delta, c_\varnothing) \right)
$$
(5)

where $c_\varnothing$ is the unconditional condition implemented by scene-level label dropout, and $s$ is the guidance scale. Given $u_\theta^{\mathrm{cfg}}$, we update latents by a large-step transport rule:

$$
z_{t-\Delta} = z_t - \Delta u_\theta^{\mathrm{cfg}}(z_t; t, \Delta, c). \quad (6)
$$

Setting $(t, \Delta) = (1, 1)$ gives solver-free one-step sampling; using a small number of intermediate times gives few-step sampling. We report the resulting quality–latency trade-off in Figure 5.

**Why the JVP Term Matters for One-Step Completion.** In our setting, a single large-step update must preserve edge-defined constraints (lane connectivity and lane–agent alignment) in one shot. The fixed-$\Delta$ JVP term calibrates the interval-velocity predictor at the boundary $(t, \Delta)=(1, 1)$, which is where one-step failures concentrate.

**Comparison with DDPM and Flow-Matching Baselines.** DDPM trains a discrete-time denoiser by regressing injected noise and typically samples by iterating many reverse steps, while rectified flow matching trains a continuous-time velocity field on the straight-line path and samples via ODE integration. VectorWorld adopts the same latent interface and edge-gated relational DiT backbone across DDPM / Flow / MeanFlow, but deploys MeanFlow because it is explicitly trained for large-step transport, enabling the solver-free update. A unified formulation and our masking/clamping implementation are detailed in Appendix A.7 (DDPM/Flow) and Appendix A.8 (MeanFlow).

### 3.5. Streaming Outpainting

**Trigger, Clamp, Complete.** VectorWorld maintains an ego-centered local tile and triggers outpainting only when the ego approaches the tile boundary. Outpainting is implemented as *joint* masked latent completion over both lane and agent tokens: tokens inside the current tile (including all observed lanes and agents) are clamped (conditioned), and the generator samples *frontier lane tokens and frontier agent tokens* for the next $64\,\mathrm{m}\times64\,\mathrm{m}$ tile using the one-step update in Equation (6). New lanes and new agents can therefore both appear at the frontier; each newly generated agent is instantiated as an interaction state (static state + motion code) so $\Delta$Sim receives a history-aligned context window. The sampled latents are decoded back to vector geometry and stitched into the global scene by an SE(2) transform; Algorithm 2 (Appendix A.13.3) summarizes the full rollout loop. For closed-loop robustness, we sample $N_{\mathrm{cand}} = 8$ outpainting candidates per trigger and keep the one with the lowest validity-violation score (lane traversability, agent–lane consistency, and static collision); see Appendix A.11 for the per-candidate latency and selection rule.

### 3.6. $\Delta$Sim: Physics-Aligned NPC Policy

Even with high-fidelity scene generation, closed-loop stability depends on feasible multi-agent actions. Small per-step kinematic violations are often negligible under short-horizon open-loop metrics, but dominate kilometer-scale rollouts via compounding effects. We propose $\Delta$Sim (Fig-ure 4), combining controllable conditioning with physics-aligned action modeling and inference-time shaping.

**One-Pass RTG Conditioning FiLM.** $\Delta$Sim uses a one-pass conditional decoder whose hidden state $h$ is modulated by FiLM (Perez et al., 2018) parameters $(\gamma, \beta)$ predicted from a soft return-to-go (RTG) embedding. Specifically, the model predicts an RTG distribution $\{p_j\}_{j=1}^{K_{\mathrm{rtg}}}$ and computes a soft expectation $E = \sum_{j=1}^{K_{\mathrm{rtg}}} p_j \,\mathrm{emb}_j$, then applies FiLM as $h \leftarrow h\odot(1+\gamma)+\beta$. This provides a differentiable control interface without requiring a second forward at inference.

**Hybrid Discrete–Continuous Actions.** We parameterize the executed action in the agent body frame as a hybrid discrete–continuous update:

$$\Delta\xi = \Delta\xi_{\hat{k}} + \alpha\,\widehat{\boldsymbol{\delta}}, \qquad \hat{k} \sim \mathrm{Cat}\big(\mathrm{softmax}(\tilde{\ell})\big), \quad (7)$$

where $\widehat{\boldsymbol{\delta}} \in \mathbb{R}^3$ is a continuous refinement head $(\Delta x, \Delta y, \Delta\theta)$, and $\alpha$ is an inference-time refinement scale.

**Inference-Time Kinematic Shaping.** For each discrete action token $k$ with anchor $\Delta\xi_k = (\Delta x_k, \Delta y_k, \Delta\theta_k)$ and current state $s_t$, we compute a differentiable one-step kinematic cost

$$C_k(s_t) = \sum_m w_m\, \phi_m(s_t, \Delta\xi_k), \qquad (8)$$

where each soft penalty $\phi_m$ measures one physical constraint (yaw rate, curvature, lateral / longitudinal acceleration, reverse motion). $C_k(s_t) \in \mathbb{R}_{\geq 0}$ is low for feasible tokens and high for implausible ones. We then shape the logits before sampling:

$$\tilde{\ell}_k = \ell_k - \lambda\, C_k(s_t). \qquad (9)$$

**Differentiable Kinematic Alignment Loss (DKAL).** Let $\mathrm{ctr}(\mathbf{x})_k = x_k - \frac{1}{K_{\mathrm{kd}}}\sum_{j=1}^{K_{\mathrm{kd}}} x_j$ denote centering over the vocabulary. Given logits $\ell\in\mathbb{R}^{K_{\mathrm{kd}}}$ and costs $C(s_t)\in\mathbb{R}^{K_{\mathrm{kd}}}$, we minimize

$$\mathcal{L}_{\mathrm{dkal}} = \big\|\mathrm{ctr}(\ell) + \lambda_{\mathrm{dkal}}\, \mathrm{ctr}(C(s_t))\big\|_2^2. \qquad (10)$$

**Why centering?** The policy should learn the *relative ranking* of actions, not an arbitrary offset. (i) A constant shift of all logits leaves the softmax unchanged, so the absolute mean is unidentifiable; (ii) likewise, a constant offset of all kinematic costs is a nuisance dimension that does not affect physical preference. Centering removes both nuisances simultaneously, aligning *higher logits with lower relative costs*, and ensures the training-time signal in Equation (10) is consistent with the inference-time shaping in Equation (9).

*Table 1.* **Quantitative comparison of scene generation quality.** We evaluate VectorWorld against prior vectorized scene generators on nuPlan and Waymo. VectorWorld achieves strong structure and safety with competitive perceptual alignment. ↓/↑ indicates lower/higher is better. **Bold** and underline denote the best and second-best results among non-privileged methods.

| Dataset | Method | Perceptual | Map Structure | | Safety |
|---|---|---|---|---|---|
| | | FD ↓ | Route Len. (m) ↑ | Endpt. Dist. (m) ↓ | Coll. Rate (%) ↓ |
| **nuPlan** | SLEDGE-L (Chitta et al., 2024) | 1.89 | $35.34_{\pm 8.63}$ | $0.470_{\pm 0.320}$ | 22.30 |
| | SLEDGE-XL (Chitta et al., 2024) | 1.44 | $35.83_{\pm 8.35}$ | $0.420_{\pm 0.290}$ | 21.20 |
| | ScenDream-B (Rowe et al., 2025) | 1.05 | $37.04_{\pm 10.21}$ | $0.320_{\pm 0.800}$ | 11.90 |
| | ScenDream-L (Rowe et al., 2025) | **0.67** | $36.87_{\pm 10.37}$ | $0.250_{\pm 0.710}$ | 9.30 |
| | VectorWorld (Ours) | 0.98 | **$37.22_{\pm 11.40}$** | **$0.078_{\pm 0.106}$** | **3.01** |
| **Waymo** | DriveSceneGen(GT)[*†] (Sun et al., 2024) | $40.59^{†}$ | $41.61_{\pm 18.61}$ | $0.010_{\pm 0.000}$ | $0.20^{*†}$ |
| | ScenDream-B (Rowe et al., 2025) | 1.61 | $38.23_{\pm 12.78}$ | $0.320_{\pm 0.900}$ | 5.40 |
| | ScenDream-L (Rowe et al., 2025) | 1.38 | $38.92_{\pm 13.56}$ | $0.210_{\pm 0.750}$ | 4.80 |
| | VectorWorld (Ours) | **0.94** | **$39.03_{\pm 15.43}$** | **$0.094_{\pm 0.483}$** | **4.69** |

[*] indicates a privileged/reference setting using ground-truth and is not ranked against learned non-privileged generators.
[†] DriveSceneGen(GT) (Sun et al., 2024) numbers on Waymo are quoted from ScenarioDreamer (Rowe et al., 2025).

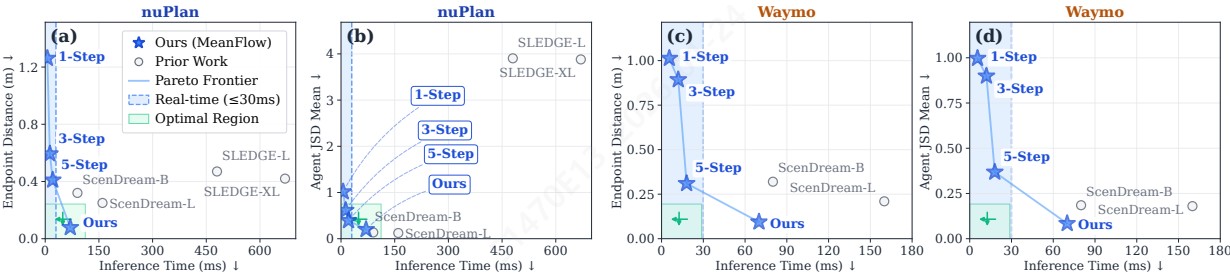

*Figure 5.* **Quality–latency trade-off for tile generation.** Endpoint distance (EPD) and mean agent-JSD are plotted against wall-clock latency as the solver-step budget varies; the vertical marker indicates the streaming budget (including one VAE decode). For the FD / Fréchet Reach / Agent-JSD numerical companion at matched latencies, Table 15 in Appendix A.15.4 provides the direct lane-graph-fidelity view.

**Closing the Loop at $k = 0$.** Finally, the interaction-state interface closes the loop: at $k = 0$, the VAE motion code $m_i$ is deterministically converted to a pseudo-history sequence that fills the NPC context window. This ensures that $\Delta$Sim receives inputs consistent with its training distribution, a prerequisite for stable rollouts when the scene is repeatedly extended by streaming outpainting.

## 4. Experiments

We evaluate VectorWorld as (i) an offline scene generator, (ii) an online outpainting engine, and (iii) a closed-loop simulator. The central question is whether a fast, one-step generator can remain structurally consistent and kinematically feasible when invoked repeatedly during rollout.

### 4.1. Setup

**Datasets and Baselines.** We use Waymo Open Motion (Gulino et al., 2023) and nuPlan (Caesar et al., 2021). We compare to vectorized scene generators SLEDGE (Chitta et al., 2024) (L/XL) and ScenDream (Rowe et al., 2025) (B/L).

**Metrics.** Formal metric definitions appear in Appendix A.12. Briefly: *EPD* (Endpoint Distance) measures mean Euclidean distance between a lane endpoint and its successor's start point; *JSD* is the Jensen–Shannon divergence between generated and ground-truth scalar-attribute histograms; *Agent-JSD mean* averages six per-attribute JSDs (nearest, lateral, angular, length, width, speed), excluding count JSD per Scenario Dreamer; and *FD* is the Fréchet distance over frozen autoencoder lane embeddings. Offline initialization quality is assessed via FD, route length, EPD, and static collision rate (Table 1), plus Urban-Planning topology FDs and agent count JSD (Table 4). For sampler comparisons (Figure 5), efficiency is end-to-end latency per $64\,\text{m} \times 64\,\text{m}$ tile (sampling + one VAE decode); the 30 ms marker denotes the online per-candidate budget. Closed-loop rollouts use $N_{\text{cand}} = 8$, yielding $\approx 44.8$ ms per trigger (Appendix A.11).

**Closed Loop.** We evaluate an ego planner interacting with reactive NPCs and report collision/offroad/success, route progress, and jerk (Table 2; definitions in Appendix A.13). We use three probes: a rule-based IDM probe (Treiber et al.,

*Table 2.* **Closed-loop evaluation on Waymo. Units:** FD: unitless; S.Coll., Coll., Offroad, Success: %; Route: m. **Columns:** *NPC Policy*: background traffic controller; *Ego Planner*: agent under test. **Key:** Warm = motion-history warm start; Online = streaming outpainting. **Note:** Scene Val. metrics are recomputed per simulator configuration after its own initialization/outpainting and candidate-selection pipeline, serving as configuration-level diagnostics rather than duplicates of Table 1.

| Simulator | NPC Policy | Ego Planner | Sys. Cap. | | Scene Val. | | Ego Performance | | | | Dyn. Metric |
|---|---|---|---|---|---|---|---|---|---|---|---|
| | | | Online | Warm | FD ↓ | S.Coll ↓ | Coll. | Off. | Succ. | Route (m) | Value |
| *I. Platform Stability & Consistency (Goal: Verify simulation realism using a standard IDM driver (Treiber et al., 2000).)* | | | | | | | | | | | |
| Waymo | Replay | IDM (Rule-based) | ✗ | ✗ | 0.00 | – | 7.2 | 5.8 | 87.0 | 64 | 11.4 (Jerk) |
| ScenDream-L | Ctrl-Sim | IDM (Rule-based) | ✗ | ✗ | 1.38 | 4.80 | 12.4 | 5.9 | 81.7 | 200 | 15.5 (Jerk) |
| VectorWorld | $\Delta$Sim | IDM (Rule-based) | ✗ | ✗ | 1.05 | 4.69 | 7.5 | 1.5 | 91.0 | 200 | 10.4 (Jerk) |
| VectorWorld | $\Delta$Sim | IDM (Rule-based) | ✓ | ✗ | 1.19 | 7.39 | 45.5 | 12.5 | 42.0 | 1000 | 16.6 (Jerk) |
| VectorWorld | $\Delta$Sim | IDM (Rule-based) | ✓ | ✓ | **0.94** | **2.95** | 18.5 | 3.5 | 78.0 | 1000 | 9.6 (Jerk) |
| *II. Failure Discovery & Stress Test (Goal: Attack a fixed PPO policy (Schulman et al., 2017) to find failures.)* | | | | | | | | | | | |
| Waymo | Replay | PPO (Pre-trained) | ✗ | ✗ | 0.00 | – | 29.3 | 6.9 | 63.8 | 200 | 36.2 (Fail %) |
| ScenDream-L | Ctrl-Sim (Pos.) | PPO (Pre-trained) | ✗ | ✗ | 1.42 | 5.10 | 52.8 | 9.1 | 38.2 | 200 | 61.8 (Fail %) |
| ScenDream-L | Ctrl-Sim (Neg.) | PPO (Pre-trained) | ✗ | ✗ | 1.42 | 11.7 | 59.0 | 9.0 | 32.1 | 200 | 67.9 (Fail %) |
| VectorWorld | $\Delta$Sim (Pos.) | PPO (Pre-trained) | ✓ | ✓ | **0.92** | **2.13** | 30.7 | 7.2 | 62.3 | 1000 | 37.9 (Fail %) |
| VectorWorld | $\Delta$Sim (Neg.) | PPO (Pre-trained) | ✓ | ✓ | 1.15 | 8.55 | **65.1** | **9.2** | **25.7** | 1000 | **75.0** (Fail %) |
| *III. Closed-Loop Training (Goal: Retrain PPO in simulator, evaluate on VectorWorld (Ours).)* | | | | | | | | | | | |
| VectorWorld | $\Delta$Sim (Neg.) | PPO (Pre-trained) | ✓ | ✓ | 1.15 | 8.55 | 65.1 | 9.2 | 25.7 | 1000 | 25.7 (Succ %) |
| VectorWorld | $\Delta$Sim (Neg.) | PPO (Retrained) | ✓ | ✓ | 1.15 | 8.55 | **35.1** | **7.9** | **56.0** | 1000 | **56.0** (Succ %) |

*Table 3.* $\Delta$Sim **ablation on Waymo.** Components: return-to-go conditioning (RTG), hybrid anchor+residual action head (A+R), and DKAL. Metrics: average displacement error (ADE; ↓), controllability rank-correlation ($\rho$; ↑), and inference cost (latency / forward passes; ↓).

| Method | Components | | | Acc. | Ctrl. | Effic. | |
|---|---|---|---|---|---|---|---|
| | 1-pass | A+R | DKAL | ADE ↓ | $\rho$ ↑ | Lat. ↓ | Fwd ↓ |
| Ctrl-Sim (Rowe et al., 2024) | ✗ | ✗ | ✗ | 2.80 | -0.25 | 36.9 | 2.0 |
| $\Delta$Sim (w/o Refine) | ✓ | ✗ | ✓ | 2.62 | 0.39 | **21.7** | **1.0** |
| $\Delta$Sim (w/o DKAL) | ✓ | ✓ | ✗ | 2.10 | 0.24 | 26.1 | **1.0** |
| $\Delta$Sim (Ours) | ✓ | ✓ | ✓ | **1.72** | **0.53** | 26.1 | **1.0** |

*Table 4.* **Urban-Planning topology and density calibration.** We report keypoint-graph Fréchet distances (Conn./Dens./Reach/Conve.) and agent count JSD, complementing Table 1. VectorWorld improves connectivity and density on nuPlan, density on Waymo, and count JSD on both datasets, indicating better local lane continuity and density calibration; the remaining Reach/Conve. gaps concentrate on minor connectors in the compact keypoint-graph evaluation rather than overall realism.

| Dataset | Method | Conn. ↓ | Dens. ↓ | Reach ↓ | Conve. ↓ | Cnt.JSD ↓ |
|---|---|---|---|---|---|---|
| nuPlan | ScenDream-L | 0.18 | 0.43 | **0.03** | **0.33** | 0.011 |
| | **VectorWorld (Ours)** | **0.09** | **0.41** | 0.09 | 0.79 | **0.004** |
| Waymo | ScenDream-L | **0.03** | 0.95 | **0.28** | **2.05** | 0.029 |
| | **VectorWorld (Ours)** | 0.23 | **0.57** | 0.37 | 2.63 | **0.001** |

2000) for platform stability, a fixed PPO policy (Schulman et al., 2017) for stress testing, and PPO retraining to measure training value.

## 4.2. Results

**Offline Scene Quality.** Table 1 shows that VectorWorld improves structural integrity and initialization validity on both datasets. On nuPlan, VectorWorld reduces EPD from 0.250 m (ScenDream-L) to 0.078 m and reduces the collision rate from 9.30% to 3.01%. On Waymo, VectorWorld achieves the lowest FD among non-privileged methods (0.94) while improving route length and EPD.

**Qualitative Comparison.** Figure 6 visualizes representative samples on both datasets. The improvements concentrate at intersections and merges: VectorWorld better preserves lane successor continuity and reduces agent–lane inconsistencies, matching the gains in endpoint distance and initialization collision rate in Table 1.

**Per-component Ablation Summary.** Table 5 consolidates how each module contributes to the deployment constraint it targets. The interaction-state VAE simultaneously lowers offline initialization collisions and closed-loop jerk, the edge-gated DiT improves route validity and endpoint distance under masked completion, the JVP + two-time MeanFlow makes single-step generation usable at the same 5.6 ms per-candidate budget (validity $9.3\% \rightarrow 87.5\%$), and $\Delta$Sim simultaneously reduces ADE and lateral-acceleration violations. Numbers in each row are reproduced from the corresponding per-module ablation in Appendix A.15.

**Real-time Efficiency–Quality Trade-off.** Figure 5 reports the quality–latency curve as we vary the solver step budget. MeanFlow-1 stays within the per-candidate real-time regime and remains competitive in EPD and mean agent-JSD, enabling event-triggered streaming outpainting; increasing steps improves quality but quickly exceeds the

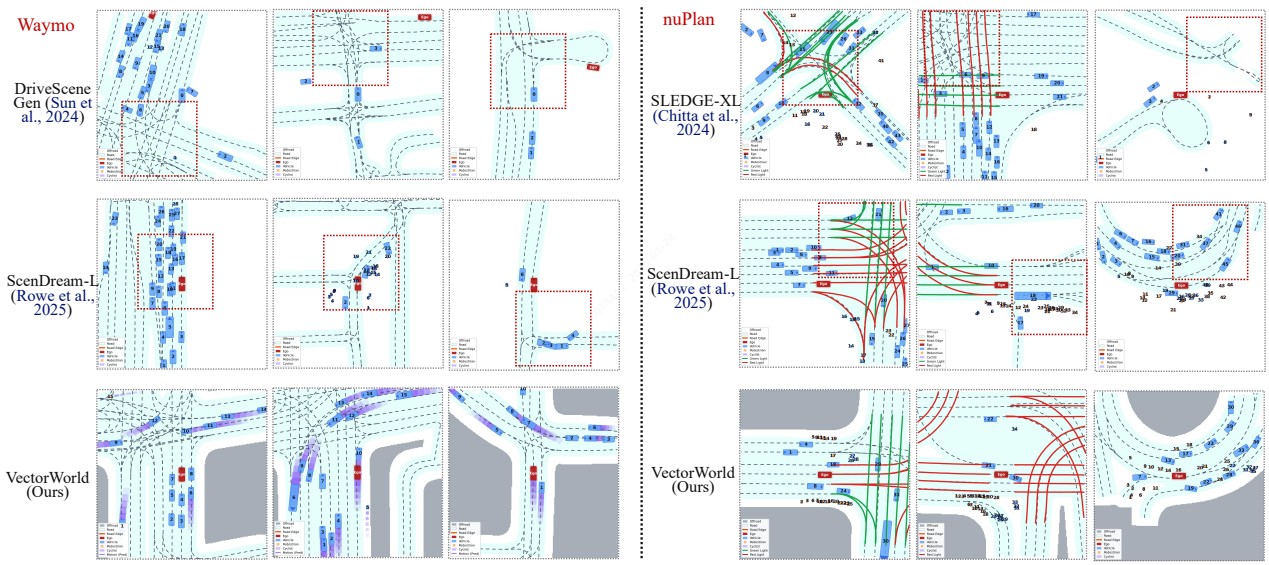

*Figure 6.* **Qualitative comparison on Waymo and nuPlan.** Visualizing decoded elements in $64\,\text{m} \times 64\,\text{m}$ tiles. VectorWorld demonstrates superior lane connectivity and fewer agent overlaps at initialization compared to prior vectorized generators.

*Table 5.* **Per-component ablation summary.** Each module is paired with the deployment constraint it primarily targets; the row reports the most diagnostic baseline vs. ours, with full per-module ablations in Appendix A.15 (Tables 11, 12, 13, 16).

| Module | Baseline | Ours | Diagnostic metric |
|---|---|---|---|
| Interaction-state VAE | 9.45 | **4.69** | Init. coll. (%)↓ |
| | 16.6 | **9.6** | Closed-loop jerk ↓ |
| Edge-gated DiT | 87.4 | **98.1** | Route validity (%)↑ |
| | 0.301 | **0.094** | EPD (m)↓ |
| MeanFlow (1-step) | 9.3 | **87.5** | Valid scenes (%)↑ |
| | 8.52 | **0.269** | EPD (m)↓ |
| ΔSim (full) | 2.80 | **1.72** | ADE (m)↓ |
| | 14.2 | **2.1** | Lat.-acc. viol. (%)↓ |

online sampler budget.

**Ablation of** $\Delta$Sim. Table 3 shows that RTG conditioning, hybrid anchor–residual actions, and DKAL are complementary. Compared to Ctrl-Sim (Rowe et al., 2024), the full $\Delta$Sim improves ADE from 2.80 m to 1.72 m and increases controllability ($\rho$) from $-0.25$ to 0.53, while using a single forward pass.

**Closed-Loop System.** Table 2 shows that online outpainting extends evaluation to $1\,\text{km}+$ but exposes compounding errors. *Why history-free initialization produces jerk spikes:* both the ego planner and $\Delta$Sim are history-conditioned and read a short fixed-length window to infer velocity, curvature, and braking intent. With a history-free start, the missing frames are fabricated (zero-padded or repeated), creating an out-of-distribution input: the current state in-

dicates motion while the pseudo-history implies the agent was static, so the first executed action is a large corrective step ($j_t = (a_t - a_{t-1})/\Delta t$ becomes a spike). Our motion-code interaction state reconstructs a dynamically consistent pseudo-history; consequently, warm-started initialization reduces jerk ($16.6 \rightarrow 9.6$) and increases success ($42.0\% \rightarrow 78.0\%$). Controllable "tilt" produces targeted stress-test distributions, and ego planner (PPO (Schulman et al., 2017)) retraining in VectorWorld increases success on the hardest setting from $25.7\%$ to $56.0\%$. At evaluation, we control NPC aggressiveness by exponentially tilting the RTG distribution with coefficient $\kappa$ (Appendix A.9).

**Additional Experiments, Limitations, and Safety.** Further setups, ablations, qualitative analyses, and discussions on limitations and safety are provided in Appendix A.

## 5. Conclusion

We introduced VectorWorld, a streaming vector-graph world model for closed-loop autonomous-driving simulation that outpaints lane–agent tiles on demand. It couples (i) a motion-aware interaction-state VAE that aligns initialization with history-conditioned policies, (ii) an edge-gated relational DiT trained with interval-conditioned MeanFlow and JVP supervision for solver-free one-step masked completion, and (iii) a physics-aligned NPC policy ($\Delta$Sim) that reduces long-horizon compounding from infeasible actions. Across Waymo Open Motion and nuPlan, VectorWorld improves initialization validity, enables $1\,\text{km}+$ closed-loop rollouts, and yields stress-test and policy-training gains.

## Impact Statement

This work aims to improve the fidelity and efficiency of learned simulation for autonomous-driving research. A reliable closed-loop simulator can reduce real-world testing risk by enabling stress testing, counterfactual evaluation, and data-efficient policy training in a controlled environment.

Potential negative impacts include misuse for generating adversarial traffic scenarios intended to exploit specific policy weaknesses, or over-reliance on a simulator that contains modeling bias. To mitigate these risks, we recommend (i) using generated scenarios only in sandboxed simulation, (ii) reporting feasibility thresholds and post-processing rules used by the backend, and (iii) validating policy improvements with cross-backend evaluation (e.g., Appendix A.15.6).

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

# A. Appendix

This appendix provides (i) a consolidated notation with tensor shapes, (ii) detailed specifications of the vectorized scene representation and model components (motion-aware VAE, edge-gated relational DiT, MeanFlow training with JVP, and $\Delta$Sim behavior model), and (iii) reproducible training and closed-loop evaluation protocols. We summarize notation in Tables 6–8 for quick lookup.

## A.1. Appendix table of contents

## A.2. Notation and Tensor Shapes

We adopt a consistent notation that disambiguates (i) latent-generation time $t \in [0, 1]$ from simulation step indices $k \in \{0, \ldots, K\}$, and (ii) attention $\mathbf{V}$ (values) from k-disks vocabulary $\mathbf{V}_{\mathrm{kd}}$. We use bold $\boldsymbol{\epsilon}$ to denote Gaussian noise and $\mathcal{E}$ to denote graph edge sets; implementation uses adjacency relations and index tensors as listed below.

## A.3. Vectorized Scene Representation

We represent a traffic scene as a heterogeneous graph with lane and agent nodes. The choice of scene representation has been examined for downstream robotic learning, with surveys cataloguing options from geometric primitives to foundation-

*Table 6.* Core notation for the vectorized scene representation (per scene unless stated otherwise).

| Symbol | Type / shape | Meaning |
|---|---|---|
| $\mathcal{G} = (\mathcal{V}, \mathcal{E})$ | set-valued | Heterogeneous lane–agent vector graph for one ego-centric tile. |
| $\mathcal{V}_{\text{lane}}, \mathcal{V}_{\text{agent}}$ | sets | Lane nodes / agent nodes in $\mathcal{G}$. |
| $N_\ell$ | integer | Number of lane segments in a scene (capped by $N_\ell^{\max}$). |
| $N_a$ | integer | Number of agents in a scene (capped by $N_a^{\max}$; includes ego). |
| $P$ | integer | Number of points per lane polyline for the generator/autoencoder ($P{=}20$ in Waymo generation). |
| $P_\Delta$ | integer | Number of points per lane polyline for $\Delta$Sim context ($P_\Delta{=}50$ in Waymo behavior). |
| $K_{\text{mot}}$ | integer | Number of points in the motion-code polyline (per agent), i.e., $m_i \in \mathbb{R}^{2K_{\text{mot}}}$. |
| $\mathbf{L}$ | $\mathbb{R}^{N_\ell \times P \times 2}$ | Lane centerlines in the ego frame, resampled to $P$ points, each point is $(x, y)$. |
| $\mathbf{S}$ | $\mathbb{R}^{N_a \times 7}$ | Static agent state: $(x, y, v, \cos\theta, \sin\theta, \ell, w)$, normalized. |
| $\mathbf{M}$ | $\mathbb{R}^{N_a \times 2K_{\text{mot}}}$ | Motion code: $K_{\text{mot}}$ body-frame polyline points per agent, $(x_1, y_1, \ldots, x_{K_{\text{mot}}}, y_{K_{\text{mot}}})$. |
| $\mathbf{c}^a$ | $\{0,1\}^{N_a \times C_a}$ | Agent type one-hot ($C_a{=}3$: vehicle, pedestrian, cyclist). |
| $\mathbf{c}^\ell$ | $\{0,1\}^{N_\ell \times C_\ell}$ | Lane semantic attributes (Waymo: dummy constant; nuPlan: may include traffic-light state when available). |
| $\mathcal{A}_{\ell\ell}$ | subset of $[N_\ell] \times [N_\ell]$ | Directed lane-to-lane adjacency relation used for message passing. |
| $\mathcal{A}_{aa}$ | subset of $[N_a] \times [N_a]$ | Agent-to-agent adjacency relation (complete in our implementation). |
| $\mathcal{A}_{\ell a}$ | subset of $[N_\ell] \times [N_a]$ | Lane-to-agent adjacency relation (bipartite complete in our implementation). |
| $\mathbf{T}_{\ell\ell}$ | $\{0,1\}^{|\mathcal{A}_{\ell\ell}| \times C_{\text{conn}}}$ | One-hot lane connection type per directed edge (Waymo: $C_{\text{conn}}{=}6$: none/pred/succ/left/right/self). |
| $\mathbf{I}_{\ell\ell}$ | $\mathbb{Z}^{2 \times |\mathcal{A}_{\ell\ell}|}$ | Edge index (source, destination) for lane-to-lane edges (PyG convention). |
| $\mathbf{I}_{aa}$ | $\mathbb{Z}^{2 \times |\mathcal{A}_{aa}|}$ | Edge index for agent-to-agent edges. |
| $\mathbf{I}_{\ell a}$ | $\mathbb{Z}^{2 \times |\mathcal{A}_{\ell a}|}$ | Edge index for lane-to-agent edges. |
| $\mathbf{m}^\ell$ | $\{0,1\}^{N_\ell}$ | Partition mask: 1 indicates "conditioned" lanes (clamped during masked completion). |
| $\mathbf{m}^a$ | $\{0,1\}^{N_a}$ | Partition mask for agents in inpainting/outpainting. |

model embeddings (Deng et al., 2025) and complementing recent reviews of interactive 3D scene generation (Li et al., 2026). We adopt a vector graph precisely because it exposes lane connectivity at the edge level, which is the structure our masked-completion interface and topology-aware metrics directly exploit. Each scene is normalized into an ego-centric SE(2) frame by translating the ego to the origin and applying a deterministic rotation such that the ego heading aligns with a fixed forward axis. We then crop to a square field-of-view (FOV) and cap the number of lanes and agents.

For Waymo generation, the autoencoder and latent generator operate on a $64\,\text{m} \times 64\,\text{m}$ FOV, with at most $N_\ell^{\max}{=}100$ lane segments and $N_a^{\max}{=}30$ agents. Lane segments are represented by $P{=}20$ resampled points.

Lane connectivity is encoded as a typed directed relation. Concretely, we construct a complete directed lane-to-lane edge set and assign each edge a connection-type one-hot $\mathbf{T}_{\ell\ell} \in \{0,1\}^{|\mathcal{A}_{\ell\ell}| \times 6}$ indicating $\{\text{none}, \text{pred}, \text{succ}, \text{left}, \text{right}, \text{self}\}$. This separates (i) the *message-passing topology* (complete) from (ii) the *semantic lane-graph structure* (typed edges), which is critical for topology metrics.

For outpainting/inpainting, we additionally construct a partitioned scene variant by splitting lane segments that cross a fixed partition line (Waymo: $y = 0$ in the ego frame). Elements on the "before-partition" side are treated as conditions. We use boolean masks to (i) prevent encoder attention across the partition and (ii) keep conditioned latents fixed during generation.

**Mask convention.** We use boolean masks where `True` indicates *conditioned / clamped* tokens that must be preserved, and `False` indicates tokens to be generated. All diffusion/flow/MeanFlow variants implement conditional completion by keeping clamped tokens fixed and setting their training targets to zero velocity/noise.

### A.4. Motion Code: Definition, Normalization, and Invariance

We augment each agent with a compact motion code that summarizes its recent history as a short polyline in the agent body frame. For each agent $i$, we define a static state $s_i \in \mathbb{R}^7$, an agent type $\tau_i$, and a motion-history code $m_i \in \mathbb{R}^{2K_{\text{mot}}}$ represented as a $K_{\text{mot}}$-point polyline in the agent frame.

*Table 7.* Notation for latent variables and generative dynamics (VAE + diffusion/flow/MeanFlow).

| Symbol | Type / shape | Meaning |
|---|---|---|
| $\text{Enc}_\phi$ | function | VAE encoder mapping a scene graph to Gaussian posterior parameters. |
| $\text{Dec}_\psi$ | function | VAE decoder mapping sampled latents to reconstructed scene elements. |
| $\mathbf{z}^\ell$ | $\mathbb{R}^{N_\ell \times d_\ell}$ | Lane latents (Waymo: $d_\ell{=}24$). |
| $\mathbf{z}^a$ | $\mathbb{R}^{N_a \times d_a}$ | Agent latents (Waymo: $d_a{=}18$). |
| $\mathbf{z}$ | $\mathbb{R}^{(\cdot)}$ | Concatenated latents, $\mathbf{z} = (\mathbf{z}^\ell, \mathbf{z}^a)$. |
| $\mathbf{z}_t$ | $\mathbb{R}^{(\cdot)}$ | Noisy latent on the rectified path at time $t$: $\mathbf{z}_t = (1-t)\mathbf{z} + t\boldsymbol{\epsilon}$. |
| $\boldsymbol{\mu}^\ell, \log\boldsymbol{\sigma}^{2,\ell}$ | $\mathbb{R}^{N_\ell \times d_\ell}$ | Lane posterior mean / log-variance from $\text{Enc}_\phi$. |
| $\boldsymbol{\mu}^a, \log\boldsymbol{\sigma}^{2,a}$ | $\mathbb{R}^{N_a \times d_a}$ | Agent posterior mean / log-variance from $\text{Enc}_\phi$. |
| $\boldsymbol{\epsilon}$ | $\mathbb{R}^{(\cdot)}$ | Standard Gaussian noise, $\boldsymbol{\epsilon} \sim \mathcal{N}(\mathbf{0}, \mathbf{I})$. |
| $t$ | scalar in $[0,1]$ | Continuous time for flow / MeanFlow (not the simulation step). |
| $r$ | scalar in $[0,1]$ | MeanFlow auxiliary time with constraint $r \le t$. |
| $\Delta$ | scalar in $[0,1]$ | MeanFlow interval length, $\Delta = t - r$ (second time channel). |
| $v^\star$ | $\mathbb{R}^{(\cdot)}$ | Rectified-flow target velocity, $v^\star = \boldsymbol{\epsilon} - \mathbf{z}$ (zero for clamped tokens). |
| $c$ | (structured) | Generation context: masks, counts, and optional scene labels. |
| $c_\varnothing$ | (structured) | Unconditional context used for CFG (implemented via label dropout). |
| $u_\theta$ | function | MeanFlow predictor: mean velocity over interval $[r, t]$ at state $\mathbf{z}_t$. |
| $V_\theta$ | function | JVP-corrected MeanFlow predictor: $V_\theta = u_\theta + \Delta \frac{\mathrm{d}}{\mathrm{d}t} u_\theta$. |
| $s$ | scalar | Classifier-free guidance (CFG) scale used at sampling time. |

Given the global trajectory up to the selected timestamp, we transform historical positions into the agent's body frame at the current time by applying the inverse SE(2) transform using the agent's current pose. We then truncate the history to the most recent $L_{\max}$ meters of arc length (Waymo default $L_{\max}{=}12$ m) and resample $K_{\text{mot}}$ points uniformly in arc length.

To explicitly capture the discrete "static" mode, we set the motion code to all zeros if the agent is deemed static. An agent is classified as static if either (i) the maximum displacement over the most recent history window is below a threshold, or (ii) the mean speed is below a threshold (see Table 9 for values). This produces a clear static cluster in latent space and stabilizes downstream generation.

We normalize the motion polyline into $[-1, 1]$ using fixed physical ranges in the body frame. For Waymo, longitudinal coordinates are mapped from $[-D, 0]$ and lateral coordinates from $[-Y, Y]$, where $D{=}12$ m and $Y{=}6$ m by default.

Importantly, the motion code is defined in the agent body frame and is therefore invariant to global SE(2) transforms used in tile stitching. When extending a simulation environment by SE(2)-transforming a new tile into a global frame, we transform agent *static states* and lane polylines, but keep the motion code unchanged.

**Relation to Dense Scene-Flow Representations.** At a conceptual level, our motion code is a task-specific, compact analog of dense 3D scene-flow representations that have been studied for perception, including unsupervised adversarial learning of point-wise motion (Wang et al., 2022), pseudo-LiDAR formulations that bridge image and LiDAR motion estimation (Jiang et al., 2022a), pseudo auto-labelling that scales supervision via geometric self-consistency (Jiang et al., 2024b), and uncertainty-aware diffusion estimators (Liu et al., 2025); beyond instantaneous flow, neural fields and Gaussian deformation fields offer a 4D continuum view of moving point clouds (Jiang et al., 2024a). Compared with these per-point dense representations, our agent-frame polyline is intentionally compact and is designed to interface directly with the discrete kinematic action heads of $\Delta$Sim, allocating capacity to the short-horizon dynamics that reactive history-conditioned policies actually consume at $t{=}0$.

### A.5. Motion-aware Variational Autoencoder

The main architecture of the motion-aware gated VAE is shown in Figure 2. This appendix provides the implementation details and the empirical gate visualization that validates the intended selective-encoding behavior.

Our scene autoencoder is a motion-aware VAE that encodes lanes, agents, and lane connectivity into Gaussian latent variables, and decodes them back to vectorized geometry and discrete attributes.

*Table 8.* Notation for the $\Delta$Sim behavior model and closed-loop simulation.

| Symbol | Type / shape | Meaning |
| --- | --- | --- |
| $\pi_{\text{ego}}$ | policy | Ego planner/policy used in closed-loop rollout. |
| $\pi_{\Delta}$ | policy | NPC policy implemented by $\Delta$Sim. |
| $R$ | scalar | Tile radius (ego-centered observation crop) used by the simulator. |
| $\tau$ | scalar | Outpainting trigger distance to the tile boundary (Algorithm 2). |
| $\mathbf{V}_{\text{kd}}$ | $\mathbb{R}^{K_{\text{kd}} \times 3}$ | k-disks vocabulary of SE(2) deltas, $(\Delta x, \Delta y, \Delta \theta)$, with $K_{\text{kd}}{=}384$. |
| $a_k$ | integer | Discrete action token index at simulation step $k$. |
| $\ell$ | $\mathbb{R}^{K_{\text{kd}}}$ | Action logits over the k-disks vocabulary before shaping. |
| $\tilde{\ell}$ | $\mathbb{R}^{K_{\text{kd}}}$ | Shaped logits, $\tilde{\ell}_k = \ell_k - \lambda C_k(s_t)$. |
| $C_k(s_t)$ | scalar | Differentiable kinematic cost of token $k$ given current state $s_t$. |
| $\lambda$ | scalar | Inference-time shaping weight. |
| $\lambda_{\text{dkal}}$ | scalar | DKAL regularization weight used in training. |
| $\alpha$ | scalar | Residual refinement scale in the hybrid action head. |
| $\widehat{\boldsymbol{\delta}}_k$ | $\mathbb{R}^3$ | Continuous residual refinement predicted by the residual head. |
| $\widehat{\Delta \mathbf{s}}_k$ | $\mathbb{R}^3$ | Final local delta action: $\widehat{\Delta \mathbf{s}}_k = \mathbf{V}_{\text{kd}}[a_k] + \widehat{\boldsymbol{\delta}}_k$. |
| $K_{\text{rtg}}$ | integer | Number of RTG bins ($K_{\text{rtg}}{=}350$ in Waymo). |
| $\{p_j\}_{j=1}^{K_{\text{rtg}}}$ | simplex | Predicted RTG distribution (used to form a soft control embedding). |
| $(\gamma, \beta)$ | vectors | FiLM modulation parameters applied to decoder hidden states. |
| $\kappa$ | scalar | Exponential tilting coefficient applied at inference for controllability sweeps. |
| $\Delta t$ | scalar | Simulation step size (Waymo closed-loop: $\Delta t{=}0.1$ s). |
| $K_{\text{sim}}$ | integer | Number of simulation steps per episode (default $K_{\text{sim}}{=}400$). |

### A.5.1. AGENT ENCODING WITH MOTION-AWARE GATING

Each agent is embedded using two branches: a static branch operating on the 7D instantaneous state and a motion branch operating on the $2K_{\text{mot}}$-dimensional motion code. A learnable gating network predicts a per-channel gate vector $g \in (0,1)^d$ from the concatenated agent input, and fuses the two representations as

$$\mathbf{h}_a = (1 - g) \odot \mathbf{h}_{\text{static}} + g \odot \mathbf{h}_{\text{motion}}.$$

This design allows the encoder to rely on motion features when informative, while remaining robust when motion codes are zeroed.

### A.5.2. FACTORIZED SCENE TRANSFORMER

We adopt a factorized attention architecture that alternates lane-to-lane attention with typed edge features, lane-to-agent attention, and agent-to-agent attention, with an explicit lane-edge update in each block. This preserves permutation equivariance within each node set while enabling typed relational reasoning for lane topology.

### A.5.3. LANE-COUNT DISTRIBUTION HEAD FOR INPAINTING

For partitioned scenes, we use a learnable query token that attends to conditioned lanes to predict a distribution over the number of lanes to generate in the outpainted region. The head outputs a categorical distribution over $\{0, \ldots, N_\ell^{\max}\}$, and is trained with cross-entropy on the ground-truth lane count in the after-partition region.

### A.5.4. TRAINING OBJECTIVE

Let $\widehat{\mathbf{S}}, \widehat{\mathbf{M}}, \widehat{\mathbf{L}}$ denote decoded static states, motion codes, and lane polylines. The total autoencoder loss is

$$\begin{aligned}
\mathcal{L}_{\text{AE}} = &\mathcal{L}_{\text{static}} + \lambda_{\text{mot}}\mathcal{L}_{\text{motion}} + \lambda_\ell \mathcal{L}_{\text{lane}} + \lambda_{\text{conn}}\mathcal{L}_{\text{conn}} + \lambda_{\text{type}}\mathcal{L}_{\text{type}} \\
&+ \beta_{\text{KL}}\mathcal{L}_{\text{KL}} + \lambda_{\text{cnt}}\mathcal{L}_{\text{count}} + \lambda_{\text{smooth}}\mathcal{L}_{\text{smooth}} + \lambda_{\text{col}}\mathcal{L}_{\text{collision}} + \lambda_{\text{end}}\mathcal{L}_{\text{endpoint}}.
\end{aligned} \tag{11}$$

Here, $\mathcal{L}_{\text{static}}$ is a weighted $L_1$ on $(x, y)$ versus other static dimensions; $\mathcal{L}_{\text{motion}}$ is an $L_1$ on motion codes with extra weight on static agents; $\mathcal{L}_{\text{lane}}$ is an $L_1$ over lane points; $\mathcal{L}_{\text{conn}}$ and $\mathcal{L}_{\text{type}}$ are cross-entropies for connectivity and types; and $\mathcal{L}_{\text{KL}}$ is the standard VAE KL term.

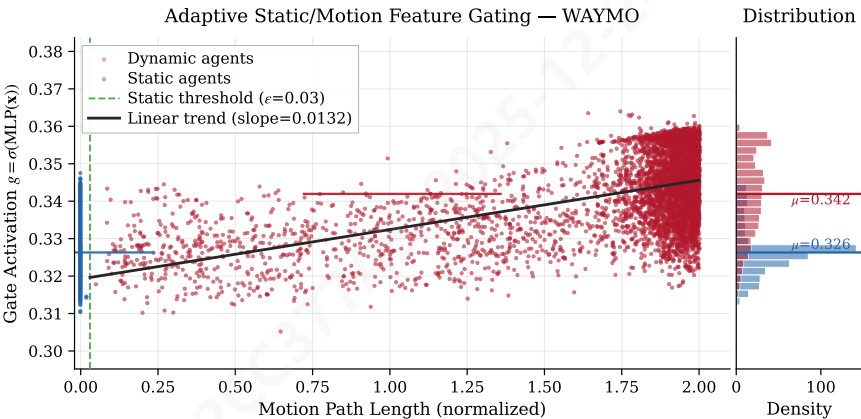

*Figure 7.* **Visualization of motion-aware gating (Waymo).** Gate activation increases with motion magnitude: static agents cluster at lower activation while dynamic agents receive higher activation, supporting the intended selective-encoding mechanism used by the interaction-state interface.

## A.6. Edge-gated relational DiT Backbone

Our latent generator uses a factorized DiT backbone that processes lane and agent latent tokens with alternating cross- and self-attention. We incorporate optional relation-aware mechanisms that target distributional metrics sensitive to relational structure (topology and agent–lane alignment). The factorized design shares spirit with heterogeneous-input fusion architectures developed for 3D perception, where projection-aware multi-modal interactions are modeled explicitly (Jiang et al., 2022b). Recent vision state-space models (Ke et al., 2026) explore adaptive structural priors as efficient alternatives to dense attention; we use typed-edge attention because it stays within our streaming latency budget while exposing the interaction structure our masked-completion interface needs, and we view such state-space alternatives as a natural extension for longer-context scenes.

### A.6.1. FACTORIZED DiT BLOCKS

Each factorized block applies, in order: agent-to-lane cross attention, lane-to-lane self attention (repeated multiple times), lane-to-agent cross attention, and agent-to-agent self attention. We allocate additional capacity to lane modeling by using multiple lane-to-lane blocks per factorized block.

### A.6.2. RELATIONAL LOGIT BIAS AND EDGE-GATED VALUES

Given token features $\mathbf{x}$, we optionally compute a per-edge per-head additive logit bias $b_{ij}^{(h)}$ by (1) projecting node features to a low-dimensional space, (2) forming pairwise edge features, and (3) mapping to head-wise biases. In addition, we optionally apply a per-edge per-head value gate $g_{ij}^{(h)} = 1 + \tanh(\cdot)$, which multiplicatively scales messages and is zero-initialized to behave as identity at the start of training.

### A.6.3. CROSS-TYPE RELATIONAL BIAS

We further support cross-type relational bias on lane-to-agent and agent-to-lane cross attention. This explicitly models agent–lane relationships and targets metrics that depend on relative geometry between agents and their nearest lanes.

### A.6.4. GLOBAL CONTEXT FUSION

Optionally, we compute a scene-level context vector by mean pooling lane and agent token features, fusing them via an MLP, and injecting the resulting delta into the conditioning stream. This provides a lightweight global pathway to capture scene-wide statistics.

## A.7. Generative Dynamics: DDPM and Rectified Flow Matching

This section summarizes the two standard objectives we build on—DDPM latent diffusion and rectified flow matching—and clarifies how conditional masked completion is implemented by clamping. MeanFlow (Section A.8) can then be read as an interval-conditioned extension designed for large-step (including one-step) transport under the same latent interface and backbone.

### A.7.1. DDPM LATENT DIFFUSION (NOISE-PREDICTION OBJECTIVE)

Let $\mathbf{z}_0$ denote a latent sample (concatenating lane and agent tokens) and let $n \in \{0, \dots, N\}$ be the discrete diffusion step. DDPM defines a forward noising process

$$\mathbf{z}_n = \sqrt{\bar{\alpha}_n}\,\mathbf{z}_0 + \sqrt{1 - \bar{\alpha}_n}\,\boldsymbol{\epsilon}, \qquad \boldsymbol{\epsilon} \sim \mathcal{N}(\mathbf{0}, \mathbf{I}), \tag{12}$$

and trains a neural denoiser $\epsilon_\theta(\mathbf{z}_n, n, c)$ to regress the injected noise:

$$\mathcal{L}_{\mathrm{ddpm}} = \mathbb{E}\Big[\big\|\epsilon_\theta(\mathbf{z}_n, n, c) - \boldsymbol{\epsilon}\big\|_2^2\Big]. \tag{13}$$

At test time, sampling requires iterating the reverse Markov chain for many steps, which is the primary latency bottleneck for streaming outpainting.

**Masked completion via clamping.** For conditioned (clamped) tokens, we do not use the DDPM forward process to perturb the observed structure. Instead, after noising the unmasked tokens, we overwrite the clamped token states with their clean values $\mathbf{z}_0$ and train the model to produce no residual update on those positions (implemented by masking their loss contribution or by using a zero supervised residual/noise target for the clamped part). This is an implementation convention for conditional completion, not an algebraic equivalence to setting $\boldsymbol{\epsilon} = \mathbf{0}$ in Equation (12). Its purpose is to ensure that conditioned structure remains fixed during inpainting/outpainting.

### A.7.2. RECTIFIED FLOW MATCHING (VELOCITY-PREDICTION OBJECTIVE)

Rectified flow matching operates in continuous time $t \in [0, 1]$ on the straight-line path between data latents and Gaussian noise:

$$\mathbf{z}_t = (1 - t)\mathbf{z}_0 + t\boldsymbol{\epsilon}. \tag{14}$$

The target velocity field is constant along the path, $v^\star = \boldsymbol{\epsilon} - \mathbf{z}_0$, and the flow-matching loss is

$$\mathcal{L}_{\mathrm{fm}} = \mathbb{E}\Big[\big\|v_\theta(\mathbf{z}_t, t, c) - (\boldsymbol{\epsilon} - \mathbf{z}_0)\big\|_2^2\Big]. \tag{15}$$

Sampling solves the ODE $\mathrm{d}\mathbf{z}_t/\mathrm{d}t = v_\theta(\mathbf{z}_t, t, c)$ from $t = 1$ to $t = 0$ using an explicit solver (e.g., Euler/Heun), which still requires multiple function evaluations for high fidelity.

**Masked completion via constant-path conditioning.** For clamped tokens, we set $\boldsymbol{\epsilon} = \mathbf{z}_0$ in Equation (14), yielding $v^\star = \mathbf{0}$. This matches the main-paper masking rule in Section 3.4 and prevents conditioned structure from drifting during inpainting/outpainting.

### A.7.3. WHY LARGE-STEP (ONE-STEP) SAMPLING IS NON-TRIVIAL

Both DDPM and flow matching are typically trained to be accurate under small reverse steps: DDPM through per-step denoising and flow matching through local ODE integration. A single-step update is therefore an extrapolation regime where small local errors become catastrophic, especially for edge-defined constraints (lane successors and agent–lane relations) that must be satisfied *in one shot*. MeanFlow addresses this by conditioning the predictor on the interval length $\Delta$ and explicitly supervising large-step transport with the identity+JVP objective (Section A.8), which is exactly the regime required by streaming outpainting.

## A.8. MeanFlow Training with Identity+JVP

We use a meanflow latent generator with an identity+JVP training objective that targets high-quality one-step sampling. The key idea is to learn an average velocity field over an interval $[r, t]$ and correct it using a Jacobian–vector product (JVP) term.

---

**Algorithm 1** MeanFlow identity+JVP training step (per minibatch).

---

1: Sample data latents $x$, sample $\epsilon \sim \mathcal{N}(\mathbf{0}, \mathbf{I})$.
2: Sample times $t \in [0, 1]$, $r \in [0, 1]$ with $r \leq t$; set $\Delta \leftarrow t - r$.
3: Form $\mathbf{z}_t \leftarrow (1 - t)x + t\epsilon$, and $\mathbf{v}_{\text{gt}} \leftarrow \epsilon - x$.
4: Compute $u_\theta(\mathbf{z}_t; t, \Delta)$, and boundary velocity $v_\theta(\mathbf{z}_t; t) \triangleq u_\theta(\mathbf{z}_t; t, \Delta{=}0)$.
5: Compute the JVP for the total derivative $\frac{\mathrm{d}}{\mathrm{d}t} u_\theta(\mathbf{z}_t; t, \Delta)$ with respect to the composite input $(\mathbf{z}_t, t, \Delta)$, using tangent $(v_\theta, 1, 0)$, i.e., $\mathrm{d}\mathbf{z}_t/\mathrm{d}t = v_\theta$, $\mathrm{d}t/\mathrm{d}t = 1$, and $\mathrm{d}\Delta/\mathrm{d}t = 0$. We stop-gradient the JVP output for stability.
6: Set $V_\theta \leftarrow u_\theta + \Delta \frac{\mathrm{d}}{\mathrm{d}t} u_\theta$.
7: Minimize $\|V_\theta - \mathbf{v}_{\text{gt}}\|_2^2$ (optionally with adaptive reweighting).

---

### A.8.1. TRAINING PATH AND GROUND-TRUTH VELOCITY

Let $x$ denote a normalized latent sample (lane and agent latents). We sample $\epsilon \sim \mathcal{N}(\mathbf{0}, \mathbf{I})$ and define the rectified path

$$\mathbf{z}_t = (1 - t)\, x + t\, \epsilon, \qquad \mathbf{v}_{\text{gt}} = \epsilon - x.$$

Conditioned (fixed) nodes use a constant path, yielding $\mathbf{v}_{\text{gt}} = \mathbf{0}$ for those nodes.

### A.8.2. IDENTITY+JVP OBJECTIVE

Let $u_\theta(\mathbf{z}_t; t, \Delta)$ denote the predicted average velocity over an interval of length $\Delta = t - r$. We also define the boundary velocity

$$v_\theta(\mathbf{z}_t; t) = u_\theta(\mathbf{z}_t; t, \Delta{=}0).$$

The identity+JVP predictor is

$$V_\theta(\mathbf{z}_t; t, \Delta) = u_\theta(\mathbf{z}_t; t, \Delta) + \Delta \left. \frac{\mathrm{d}}{\mathrm{d}t} \right|_\Delta u_\theta(\mathbf{z}_t; t, \Delta),$$

where the derivative is taken with fixed interval length $\Delta$. In implementation, the JVP is computed with respect to the composite input $(\mathbf{z}_t, t, \Delta)$ using tangent $(v_\theta, 1, 0)$. Thus the derivative includes the implicit dependence on $\mathbf{z}_t$ and the explicit dependence on $t$, while keeping the step-size channel fixed. We stop-gradient the JVP output for stability and minimize

$$\mathcal{L}_{\text{MF}} = \mathbb{E}\left[ \|V_\theta(\mathbf{z}_t; t, \Delta) - \mathbf{v}_{\text{gt}}\|_2^2 \right],$$

with optional scene-level adaptive reweighting $w = (\mathcal{L}_{\text{MF}} + c)^{-p}$.

**Sanity Checks.** When $\Delta = 0$, the correction term vanishes and the objective reduces to the flow-matching limit. When the learned velocity field is locally constant along the trajectory, $\frac{\mathrm{d}}{\mathrm{d}t}\big|_\Delta u_\theta \approx 0$ and $V_\theta \approx u_\theta$, so the JVP term does not bias training in the easy regime but mainly improves calibration for large-step transport.

### A.8.3. TIME SAMPLING

We sample $t$ and $r$ from a logit-normal distribution, enforce $r \leq t$, and set a fraction of samples to $r = t$ to retain a flow-matching limit. We also explicitly include a subset of samples with $(t = 1, r = 0)$ to cover the strict one-step regime.

### A.8.4. SAMPLING (ONE-STEP AND FEW-STEP)

At inference, we sample from $\mathbf{z}_{t=1} \sim \mathcal{N}(\mathbf{0}, \mathbf{I})$ and integrate from $t = 1$ to $t = 0$ using $K$ steps. For step $k$, with $t_{\text{cur}} > t_{\text{next}}$, we update

$$\mathbf{z}_{t_{\text{next}}} = \mathbf{z}_{t_{\text{cur}}} - (t_{\text{cur}} - t_{\text{next}})\, u_\theta(\mathbf{z}_{t_{\text{cur}}}; r = t_{\text{next}}, t = t_{\text{cur}}),$$

with classifier-free guidance applied by combining conditional and unconditional predictions.

### A.8.5. WHY INTERVAL-CONDITIONED TRAINING MATTERS?

**Why Interval-Conditioned Training Is the Right Operating Mode.** Diffusion priors are increasingly used as structured generative regularizers for downstream tasks (Lv & Jiang, 2026), and accuracy–efficiency trade-offs have been studied through knowledge distillation for spatiotemporal forecasting (Li et al., 2025). Interval-conditioned MeanFlow training pursues the same goal in the flow regime: a single backbone is trained to cover the entire interval $\Delta \in [0, 1]$, and the JVP correction calibrates the boundary at $(t, \Delta) = (1, 1)$. This makes the one-step extreme we deploy in streaming a *calibrated operating point of training* rather than a post-hoc compression of a multi-step model, which is the property required when generation is invoked repeatedly during closed-loop rollout.

### A.9. $\Delta$Sim Behavior Model (Return-conditioned Transformer)

We use a return-conditioned transformer behavior model to simulate non-ego agents in closed loop. The model predicts a discrete SE(2) action token from a k-disks vocabulary and optionally refines it with a continuous residual head, enabling high precision without requiring a prohibitively large vocabulary.

#### A.9.1. K-DISKS ACTION TOKENIZATION

We discretize local SE(2) transitions into a vocabulary $\mathbf{V}_{\mathrm{kd}} \in \mathbb{R}^{384 \times 3}$. Each entry represents a local $(\Delta x, \Delta y, \Delta\theta)$ in the agent body frame. Ground-truth actions are tokenized by minimizing a box-corner alignment error between the transformed bounding box under candidate tokens and the observed next state.

#### A.9.2. RETURN-TO-GO (RTG) MODELING AND CONTROL

We model a discounted return-to-go over a fixed horizon, discretize it into bins, and condition action prediction on RTG tokens. At inference, we optionally apply exponential tilting with coefficient $\kappa$ to shift the RTG distribution, enabling controllable agent behaviors in closed-loop simulation.

#### A.9.3. HYBRID DISCRETE–CONTINUOUS ACTION HEAD

Let $a_k$ be the predicted token and $\widehat{\boldsymbol{\delta}}_k \in \mathbb{R}^3$ be the residual refinement. The final action is

$$\widehat{\Delta \mathbf{s}}_k = \mathbf{V}_{\mathrm{kd}}[a_k] + \alpha \widehat{\boldsymbol{\delta}}_k.$$

We supervise the residual by the target $\boldsymbol{\delta}_k^\star = \Delta \mathbf{s}_k^\star - \mathbf{V}_{\mathrm{kd}}[a_k^\star]$, where $\Delta \mathbf{s}_k^\star$ is the ground-truth local transition and $a_k^\star$ is the ground-truth token index.

#### A.9.4. DKAL: PHYSICS-ALIGNED LOGIT REGULARIZATION

We optionally regularize action logits using a differentiable kinematics cost matrix computed over all vocabulary actions. The DKAL loss penalizes mismatch between centered logits and centered costs, improving physical plausibility while maintaining diversity.

### A.10. Simulation Environment Generation and Post-processing

We generate long-horizon simulation environments by iteratively outpainting tiles along a sampled route. Each tile is a vectorized scene in a fixed FOV. After generating an initial tile, we sample a route on the generated lane graph and repeatedly inpaint the next tile conditioned on the boundary region, transforming the new tile into a global frame aligned with the current route endpoint.

#### A.10.1. VALIDITY FILTERS AND POST-PROCESSING

We apply deterministic validity filters to ensure simulator compatibility: (i) lane de-duplication based on connectivity signatures and endpoint proximity; (ii) removal of overlapping agents using oriented box collision checks; (iii) removal of offroad vehicles by nearest-lane distance thresholds and heading consistency constraints; (iv) lane graph compaction by merging degree-2 chains and resampling to a fixed number of points for behavior modeling; (v) route lane-index extraction when a valid path exists (otherwise the environment is retained with a null route-lane index).

## A.11. Multi-Candidate Frontier Outpainting and Batch Latency

**Why multi-candidate sampling.**    SLEDGE (Chitta et al., 2024) and Scenario Dreamer (Rowe et al., 2025) both sample 8 inpainting candidates per step and select one via validity filtering, since a single sampled output may violate topology or feasibility constraints. To keep robustness comparisons matched while reducing inference cost, VectorWorld uses the same multi-candidate protocol in closed-loop rollout.

**Per-tile candidate budget and selection rule.**    At each outpainting trigger, we draw $N_{\text{cand}} = 8$ independent Gaussian noise samples and apply one MeanFlow update plus one VAE decode to each candidate, producing 8 candidate $64\,\text{m} \times 64\,\text{m}$ tiles. We select the candidate that minimizes a deterministic validity score combining (i) lane-graph traversability, measured by the existence of at least one outgoing successor at the frontier, (ii) agent–lane geometric consistency, measured by lateral and angular deviation under fixed thresholds, and (iii) static collision rate of newly inserted agents. The selection is fully programmatic and uses no learned re-ranker.

**Latency protocol.**    A single candidate takes $\approx 5.6\,\text{ms}$, including one MeanFlow forward and one VAE decode. With $N_{\text{cand}} = 8$ candidates, the batched closed-loop cost is $\approx 44.8\,\text{ms}$ per event-triggered outpainting update on a single H20 GPU. This is the per-trigger latency used in the closed-loop simulator. The quality–latency plots report per-candidate latency because they compare sampler operating points under a fixed candidate budget. Under the same 8-candidate protocol, multi-step diffusion baselines such as DDPM-100 or 12-step rectified flow require $\approx 1{,}280\,\text{ms}$ per trigger, so VectorWorld is $\sim 28\times$ faster while using the same robustness budget.

## A.12. Evaluation Metrics

This section defines all metrics reported in the main paper and appendix. We organize metrics by the deployment stages of VectorWorld to keep the causal chain explicit: (i) offline initial-scene quality, (ii) behavior-model quality and controllability, and (iii) closed-loop system performance under long-horizon interaction.

### A.12.1. Common preprocessing for offline metrics

**Unified Representation for Metric Computation.**    For offline evaluation, each generated scene is converted into a compact lane–vehicle representation $\{\mathcal{G}, \mathbf{L}, \mathbf{V}\}$: (i) a directed lane-successor graph $\mathcal{G}$, (ii) lane centerlines $\mathbf{L} \in \mathbb{R}^{N_\ell \times P \times 2}$ in the ego frame, and (iii) vehicle states $\mathbf{V} \in \mathbb{R}^{N_v \times 7}$ in the unified format $[x, y, v, \cos\theta, \sin\theta, \ell, w]$. To reduce sensitivity to degree-2 chains, we compact lane graphs by merging deterministic successor chains and resample each resulting centerline to a fixed number of points. This aligns the evaluation with the compact lane-graph interface used by the simulator and avoids over-penalizing tokenization artifacts.

### A.12.2. Offline initial-scene generation metrics

**Embedding-based Fréchet Distance (FD).**    FD measures distribution alignment in a fixed embedding space. We extract one embedding vector per scene by mean-pooling lane embeddings from a frozen autoencoder encoder, yielding $\mathbf{e} \in \mathbb{R}^d$ per scene. Given generated embeddings with mean/covariance $(\boldsymbol{\mu}_g, \boldsymbol{\Sigma}_g)$ and ground-truth embeddings $(\boldsymbol{\mu}_r, \boldsymbol{\Sigma}_r)$, we compute

$$D_{\text{F}} = \|\boldsymbol{\mu}_g - \boldsymbol{\mu}_r\|_2^2 + \text{Tr}\Big(\boldsymbol{\Sigma}_g + \boldsymbol{\Sigma}_r - 2(\boldsymbol{\Sigma}_g\boldsymbol{\Sigma}_r)^{1/2}\Big), \qquad \text{FD} = \sqrt{\max(D_{\text{F}}, 0)}.$$

FD is reported in Tables 1 and 2. Unlike topology metrics below, this FD is computed in the learned embedding space and is primarily a perceptual proxy.

**Route Length.**    Route length probes long-range traversability under the generated lane-successor graph. For each scene, we select an ego-proximal lane $i^\star$ whose polyline has the smallest point-to-origin distance, and a start index $s^\star$ as the closest point on that lane to the origin. For every node $j$ reachable from $i^\star$ in $\mathcal{G}$, we take the shortest-path lane sequence $\pi(i^\star \to j)$ and compute a path length by summing polyline arc lengths, using the truncated first lane segment from $s^\star$ and full lengths thereafter. The scene route length is the maximum such length over reachable nodes. We report mean and standard deviation across scenes (Tables 1 and 12).

**Endpoint Distance (EPD).**    EPD measures geometric consistency at lane transitions. For each directed successor edge $(i \to j) \in \mathcal{G}$, we compute the Euclidean distance between the end point of lane $i$ and the start point of lane $j$. We report the

mean (and optionally standard deviation) over all successor edges in all scenes.

**Static Collision Rate (Initialization Collision).** Static collision rate measures instantaneous infeasibility at $t = 0$ and is reported as a percentage. For each vehicle, we approximate its oriented box by a small set of circles whose centers lie on the longitudinal axis. A vehicle is counted as colliding if any of its circles overlaps with any circle of another vehicle. The metric is the fraction of vehicles involved in at least one collision, aggregated over all scenes. This metric is reported as Coll. Rate in Table 1 and as S.Coll in Table 2.

**Urban-Planning Topology Fréchet Distances.** Topology metrics evaluate the lane-graph structure using a keypoint graph. We construct a keypoint graph $\mathcal{K}$ by representing each lane as an edge from its start keypoint to its end keypoint, weighted by lane arc length, and merging keypoints connected by successor relations into equivalence classes. From $\mathcal{K}$, we extract four 1D feature sets: (i) keypoint degree distribution (connectivity), (ii) number of keypoints per scene (density), (iii) reachability counts per keypoint (reach), and (iv) weighted shortest-path lengths between all reachable keypoint pairs (convenience). For each feature set, we compute a 1D Fréchet distance between generated and ground-truth samples by fitting Gaussians using empirical mean and variance, and apply dataset-standard scaling factors for readability (connectivity and convenience are scaled by 10 as in prior work). These correspond to Conn., Dens., Reach, and Conve. in Table 10.

**Agent Distributional JSD Metrics.** Agent metrics quantify distribution alignment of generated vehicles relative to ground truth. For a scalar feature $x$, we compute Jensen–Shannon divergence between generated and ground-truth histograms:

$$\text{JSD}(P,Q) = \frac{1}{2}\text{KL}(P\|M) + \frac{1}{2}\text{KL}(Q\|M), \qquad M = \frac{P+Q}{2},$$

where $P, Q$ are normalized histograms over fixed bins (clipping outliers to a fixed range). In our implementation, we compute JSD as the square of the Jensen–Shannon distance returned by a standard library, so the reported value is JSD (not its square root). We report the following features, using fixed clipping ranges and bin sizes: (i) nearest-neighbor distance (all vehicles), (ii) lateral deviation to the nearest centerline point (on-road vehicles only), (iii) angular deviation from the nearest centerline heading (on-road vehicles only), (iv) length, (v) width, and (vi) speed. For readability, we scale nearest-distance and lateral-deviation JSD by 10, and angular/length/width/speed JSD by 100, matching Table 10.

**Agent Count JSD and Agent-JSD Mean.** When reported, the count JSD compares the distribution of the number of vehicles per scene between generated and ground truth using the same histogram-based JSD procedure. For efficiency plots and step-budget tables (Figure 5 and Table 14), we also report an agent-JSD mean computed as the arithmetic mean of the six scaled agent JSD metrics above (nearest, lateral, angular, length, width, speed). This produces a single scalar summary while preserving consistent units across methods.

**Validity Rate.** Validity rate (Valid %) reports the fraction of generated scenes that remain usable after deterministic decoding and validity filtering required by the simulator backend (e.g., non-empty lane set, extractable route graph, and no invalid numerical values). This metric is reported in Tables 12 and 13 and is intended to quantify failure modes that are not well captured by smooth distributional distances (e.g., catastrophic topology collapse).

**Route validity (%).** Route validity reports the fraction of scenes for which a valid route (or route-graph) can be extracted by the same deterministic procedure used by the closed-loop simulator (Appendix A.10).

### A.12.3. BEHAVIOR-MODEL METRICS

**Trajectory Accuracy (ADE).** ADE measures the average displacement error between simulated and ground-truth trajectories over a fixed future horizon. For each evaluated agent $i$ and each future time index $t \in \{1, \ldots, H\}$, let $\widehat{\mathbf{p}}_{i,t} \in \mathbb{R}^2$ be the simulated position and $\mathbf{p}_{i,t}$ the ground-truth position in the same coordinate frame. Then

$$\text{ADE} = \frac{1}{|\mathcal{I}|H} \sum_{i \in \mathcal{I}} \sum_{t=1}^{H} \|\widehat{\mathbf{p}}_{i,t} - \mathbf{p}_{i,t}\|_2,$$

where $\mathcal{I}$ is the set of evaluated agents (typically filtered to moving agents). We compute ADE under closed-loop rollout (model predictions are fed back as the next state) to reflect error accumulation.

**Controllability Correlation ($\rho$).** To quantify controllability, we run a tilt sweep over a set of control values $\kappa$ and measure a monotone behavioral response (e.g., average speed of controlled agents over the rollout). We report the Spearman rank correlation $\rho$ between $\kappa$ and the measured response, aggregated across scenes. A higher $\rho$ indicates that the control input induces a consistent monotone change in behavior.

**Closed-loop Feasibility Violation Rates.** We report per-step violation rates computed from the executed discrete action tokens and the current state. Given a k-disks token $(\Delta x, \Delta y, \Delta \theta)$ and timestep $\Delta t$, we compute derived quantities: yaw-rate magnitude $|\Delta \theta|/\Delta t$, curvature magnitude $\left|2 \sin(\Delta \theta/2)\right|/\sqrt{\Delta x^2 + \Delta y^2}$, and lateral acceleration proxy $a_{\mathrm{lat}} = v^2 \kappa$, where $v$ is a conservative reference speed. A violation is counted if any derived quantity exceeds a configured threshold. These rates are reported in Table 16.

**Efficiency (Latency and Forward Passes).** Latency is the average wall-clock time per simulation step (or per scene generation call, depending on the table), measured with GPU synchronization under a fixed batch size and precision setting. Forward passes count the number of behavior-model forward calls required per step. This distinction is crucial for return-conditioned policies: a two-pass design evaluates the network once to sample return tokens and a second time to sample actions conditioned on sampled returns, whereas our one-pass design keeps the entire decision in a single forward call.

### A.12.4. CLOSED-LOOP SYSTEM METRICS

**Episode-Level Outcomes (collision, off-route, success).** In closed-loop simulation, an episode terminates upon any of the following events: (i) ego collision with any active agent, (ii) ego deviates from the provided route by more than a fixed threshold, (iii) ego completes the route, or (iv) timeout. We report collision, off-route, and success as per-episode indicator averages over all episodes.

**Route Progress (m).** Route progress is the traveled progress of the ego along the provided route polyline at termination, reported in meters and averaged over episodes. This metric differentiates early termination (low progress) from near-completion failures (high progress but no success).

**Ego Jerk (dynamic stability proxy).** Ego jerk is computed from the ego speed time series as the time derivative of acceleration. We report a high-percentile statistic (p95 of absolute jerk) to reflect worst-case transients that impact planner stability. This metric is used as the dynamic stability value reported in Table 2 (Section I).

**Failure Rate.** When reported, failure rate is computed as $1-$success rate under the same evaluation protocol. We use it in stress-test settings to directly quantify failure discovery.

## A.13. Closed-loop Evaluation Protocol and Metrics

We evaluate planners in closed loop in generated environments. Each episode runs for $K=400$ steps with $\Delta t=0.1$ s. Non-ego agents are simulated by the $\Delta$Sim behavior model within an $80$ m $\times$ $80$ m agent FOV around the ego.

### A.13.1. EPISODE TERMINATION AND SUCCESS CRITERIA

An episode is considered successful if the ego follows the provided route without collisions and without exceeding the route-deviation threshold, within the time budget. Collisions include ego–agent collisions and (optionally) ego–map collisions under the simulator collision checker.

### A.13.2. METRIC DEFINITIONS (HIGH-LEVEL)

We report a standard set of closed-loop metrics: (i) collision rate: fraction of episodes with any ego collision; (ii) offroad rate: fraction of steps where ego exceeds a lane-distance threshold; (iii) safety margin: time-to-collision (TTC) violation rate under a threshold; (iv) physics violation rates: yaw-rate/curvature/acceleration exceedances under configured limits; (v) progress: normalized route progress and completion rate.

### A.13.3. STREAMING CLOSED-LOOP ROLLOUT (ALGORITHMIC SUMMARY)

---

**Algorithm 2** Streaming closed-loop rollout with on-demand outpainting

---

**Require:** VAE encoder/decoder $\mathrm{Enc}, \mathrm{Dec}$; MeanFlow generator $u_\theta$; condition builder $c(\cdot)$; ego planner $\pi_{\mathrm{ego}}$; NPC policy
$\quad \pi_\Delta$; outpainting trigger $\tau$; termination predicate $\mathrm{Term}(\cdot)$.
1: Sample $\boldsymbol{\epsilon} \sim \mathcal{N}(\mathbf{0}, \mathbf{I})$
2: $z \leftarrow \boldsymbol{\epsilon} - u_\theta(\boldsymbol{\epsilon}; \, t{=}1, \Delta{=}1, c_{\mathrm{init}})$
3: $\mathcal{W} \leftarrow \mathrm{Dec}(z); \mathrm{WarmStart}(\mathcal{W})$
4: $k \leftarrow 0$
5: **while** $\neg \mathrm{Term}(\mathcal{W})$ **do**
6: $\quad o_k \leftarrow \mathrm{Obs}(\mathcal{W})$
7: $\quad \hat{\tau}_k^{\mathrm{ego}} \leftarrow \pi_{\mathrm{ego}}(o_k)$
8: $\quad a_k^{\mathrm{npc}} \leftarrow \pi_\Delta(o_k)$
9: $\quad \mathcal{W} \leftarrow \mathrm{Step}(\mathcal{W}, \hat{\tau}_k^{\mathrm{ego}}, a_k^{\mathrm{npc}})$
10: $\quad$ **if** $\mathrm{NeedOutpaint}(\hat{\tau}_k^{\mathrm{ego}}, \mathcal{W}, \tau)$ **then**
11: $\quad\quad \mathcal{T}_{k+1} \leftarrow \mathrm{NextTile}(\mathcal{W}, \hat{\tau}_k^{\mathrm{ego}})$
12: $\quad\quad z_{\mathrm{obs}} \leftarrow \mathrm{Enc}(\mathrm{Clamp}(\mathcal{T}_{k+1}))$
13: $\quad\quad$ Sample $\boldsymbol{\epsilon} \sim \mathcal{N}(\mathbf{0}, \mathbf{I})$
14: $\quad\quad z_{\mathrm{new}} \leftarrow \boldsymbol{\epsilon} - u_\theta\big(\boldsymbol{\epsilon}; \, t{=}1, \Delta{=}1, c(\mathcal{T}_{k+1}, z_{\mathrm{obs}})\big)$
15: $\quad\quad \mathcal{W} \leftarrow \mathrm{Stitch}\big(\mathcal{W}, \mathrm{Dec}\big(\mathrm{Merge}(z_{\mathrm{obs}}, z_{\mathrm{new}})\big)\big)$
16: $\quad$ **end if**
17: $\quad k \leftarrow k + 1$
18: **end while**

---

## A.14. Training Setup and Hyperparameters (Waymo)

All models are trained on NVIDIA H20 GPUs. Unless otherwise noted, we use AdamW optimization with gradient clipping and deterministic seeding. The latent generator is trained with distributed data parallelism across 8 GPUs, $\Delta$Sim across 4 GPUs, and the autoencoder across 8 GPUs.

For meanflow training, we sample $t, r \in [0, 1]$ from a logit-normal distribution and enforce $r \leq t$. We set a fixed fraction of samples to $r = t$ to preserve a flow-matching limit, and include explicit $(t = 1, r = 0)$ samples to cover strict one-step training. We use an adaptive weighting $w = (\mathcal{L} + c)^{-p}$ with $p = 0.8$ and $c = 10^{-3}$ at the scene level.

For motion code extraction, we use a recent history window to classify static agents (default thresholds: maximum displacement $< 0.5$ m or mean speed $< 0.2$ m/s). We normalize longitudinal body-frame coordinates using a maximum displacement of 12 m and lateral coordinates using 6 m.

## A.15. Additional Experiments and Ablations

This section provides controlled ablations and diagnostic experiments to validate the three deployment-critical claims of VectorWorld: (i) policy-aligned initialization via interaction states, (ii) solver-free streaming generation via large-step meanflow transport on relational graphs, and (iii) long-horizon stability via physics-aligned behavior modeling. To avoid confounding factors, all generator-side ablations share the same autoencoder latent space and identical decoding and filtering rules, and all behavior-side ablations share the same k-disks vocabulary, evaluation scenes, and rollout horizon. We present ablations in the same order as the runtime pipeline so that each table can be mapped to a specific failure mode and a specific design fix: interface $\rightarrow$ generator backbone $\rightarrow$ generative dynamics $\rightarrow$ behavior model $\rightarrow$ closed-loop system value.

### A.15.1. TOPOLOGY AND DENSITY DIAGNOSTICS

Table 10 shows that VectorWorld achieves the lowest agent-count JSD on both datasets. On nuPlan, it improves lane-graph connectivity and density; on Waymo, it improves density and several agent-distribution metrics, while Conn./Reach/Conve. remain behind ScenDream-L. Thus, these diagnostics support our claims on local continuity and density calibration while explicitly recording the remaining topology gaps.

*Table 9.* Training hyperparameters on Waymo. Values are reported for the default setting used in our main experiments.

| Hyperparameter | Motion-aware VAE | MeanFlow latent generator | $\Delta$Sim behavior model |
|---|---|---|---|
| Optimizer | AdamW | AdamW | AdamW |
| Base learning rate | $1.0 \times 10^{-4}$ | $1.0 \times 10^{-4}$ | $6.0 \times 10^{-5}$ |
| $\beta_1, \beta_2$ | $(0.9, 0.999)$ | $(0.9, 0.999)$ | $(0.9, 0.999)$ |
| $\epsilon$ | $1.0 \times 10^{-7}$ | $1.0 \times 10^{-7}$ | $1.0 \times 10^{-7}$ |
| Weight decay | $2.0 \times 10^{-5}$ | $1.0 \times 10^{-5}$ | $1.0 \times 10^{-4}$ |
| Training length | 60,000 steps | 100,000 steps | 10,000 steps |
| LR schedule | linear decay | constant | linear decay |
| Warmup | 500 steps | 500 steps | 500 steps |
| Gradient clip | 10.0 | 10.0 | 5.0 |
| EMA | none | 0.9999 | none |
| Precision (training) | FP32 | FP32 | FP32 |
| GPUs | 8 (DDP) | 8 (DDP) | 4 (DDP) |
| CFG label dropout | None | 0.1 | None |
| CFG scale (sampling) | None | 4.0 | None |
| MeanFlow mode | None | identity+JVP | None |
| Two-time channel | None | enabled | None |
| RTG bins | None | None | 350 |
| RTG discount | None | None | 0.97 |
| RTG horizon | None | None | 50 steps |
| Vocabulary size | None | None | 384 |
| Residual refine | None | None | enabled |
| DKAL | None | None | enabled |

### A.15.2. MOTION-AWARE GATED VAE: INTERFACE ABLATIONS

In this section, we provide detailed ablations to validate the interaction-state interface. We focus on whether the simulator initialization provides downstream history-conditioned policies with a compatible temporal context, rather than only a static snapshot.

**Motion history is necessary but not sufficient.** The snapshot-only interface increases the initialization collision rate (9.45) and degrades endpoint distance (0.120), indicating that a history-free interface forces the downstream pipeline to infer dynamics from an implicit near-zero prior. Adding a motion code without gating reduces collisions (4.92), supporting the necessity of exposing short-horizon dynamics at initialization.

**Why naive concatenation is suboptimal.** Although concatenating motion improves downstream validity, it increases the static reconstruction error (0.017 versus 0.012). This pattern is consistent with a modality-interference failure mode: when static agents have near-zero motion, regressively modeling motion as an always-present signal injects spurious variation into static state reconstruction, which can manifest as overlapping boxes at $t = 0$.

**Selective gating improves both fidelity and validity.** The motion-aware gated variant improves static reconstruction (0.008) and motion reconstruction (0.038) while achieving the lowest endpoint distance (0.094) and collision rate (4.69). This supports the intended mechanism of the learned gate: it routes capacity toward motion only when informative and suppresses motion noise when motion codes are near-zero, which is precisely the condition that dominates parked or slow-moving agents. Importantly, this improves the interface without requiring additional sampling steps or post-hoc filtering.

**Relational encoding is not optional.** Removing the factorized scene transformer causes a collapse across all metrics (static L1 0.331; endpoint distance 0.452; collisions 13.30), showing that the interface must jointly encode lane topology and agent layout rather than treating agents independently. This aligns with the simulator deployment constraint: downstream feasibility and stability depend on edge-level consistency (lane connectivity and agent–lane relations), which cannot be recovered by a per-node encoder.

*Table 10.* **Evaluation of map topology and agent distribution alignment.** We report lane-graph **Topology** discrepancies (Conn., Dens., Reach, Conve.; lower is better) and **Agent Distributional JSD** for key attributes (lower is better). These metrics diagnose structural and density drifts that can be amplified under repeated inpainting/outpainting. **Bold** and underline denote the best and second-best results among non-privileged methods (excluding methods marked with [*]).

| Dataset | Method | Topology ↓ | | | | Agent Distributional JSD ↓ | | | | | | |
|---|---|---|---|---|---|---|---|---|---|---|---|---|
| | | Conn. | Dens. | Reach | Conve. | Near. | Lat. | Ang. | Len. | Wid. | Speed | Count |
| nuPlan | SLEDGE-L (Chitta et al., 2024) | 2.34 | 2.43 | 0.70 | 2.44 | 0.53 | 0.63 | 3.60 | 11.84 | 10.16 | 0.46 | 0.109 |
| | SLEDGE-XL (Chitta et al., 2024) | 1.67 | 1.74 | 0.51 | 1.68 | 0.49 | 0.49 | 3.26 | 11.16 | 10.29 | 0.47 | 0.217 |
| | ScenDream-B (Rowe et al., 2025) | 0.28 | 0.45 | 0.07 | **0.14** | 0.12 | 0.15 | **0.17** | **0.22** | 0.14 | 0.07 | 0.023 |
| | ScenDream-L (Rowe et al., 2025) | 0.18 | 0.43 | **0.03** | 0.33 | 0.09 | 0.11 | 0.18 | 0.25 | 0.17 | **0.06** | 0.011 |
| | Ours | **0.09** | **0.41** | 0.09 | 0.79 | 0.26 | 0.31 | 0.18 | 0.35 | 0.22 | 0.07 | **0.004** |
| Waymo | DriveSceneGen[*] (Sun et al., 2024) | 4.53 | 1.18 | 0.64 | 5.58 | 0.63 | 1.01 | 2.43 | 58.86 | 54.51 | 18.70 | 0.638 |
| | ScenDream-B (Rowe et al., 2025) | 0.17 | 1.05 | 0.56 | 3.81 | 0.07 | 0.07 | 0.07 | 0.40 | 0.25 | 0.36 | 0.072 |
| | ScenDream-L (Rowe et al., 2025) | **0.03** | 0.95 | **0.28** | **2.05** | **0.06** | **0.05** | 0.07 | 0.42 | 0.25 | 0.38 | 0.029 |
| | Ours | 0.23 | **0.57** | 0.37 | 2.63 | 0.07 | 0.07 | **0.06** | **0.13** | **0.11** | **0.15** | **0.001** |

*Table 11.* **Ablations of motion-aware gated VAE (Waymo).** We evaluate the impact of the interaction-state interface on autoencoding fidelity and downstream initialization. *Snapshot-only* lacks motion history and therefore induces cold-start artifacts. *Concat motion* exposes motion history but injects unnecessary motion noise into static agents. Our **motion-aware gated VAE** uses motion-aware gating, factorized relational encoding, and a lane-count prior, achieving the best downstream initialization validity. Lower is better for all metrics.

| VAE variant | Interface components | | | | Autoencoding | | | Downstream init. | |
|---|---|---|---|---|---|---|---|---|---|
| | Motion code | Motion gate | Factorized scene tr. | Count prior | Static L1 ↓ | Motion L1 ↓ | KL ↓ | EPD (m) ↓ | Init coll. (%) ↓ |
| Snapshot-only | ✗ | ✗ | ✓ | ✓ | 0.012 | – | 5.23 | 0.120 | 9.45 |
| Concat motion (no gate) | ✓ | ✗ | ✓ | ✓ | 0.017 | 0.045 | 4.99 | 0.102 | 4.92 |
| motion-aware gated VAE (ours) | ✓ | ✓ | ✓ | ✓ | **0.008** | **0.038** | **4.37** | **0.094** | **4.69** |
| *Additional interface checks* | | | | | | | | | |
| Ours w/o factorized transformer | ✓ | ✓ | ✗ | ✓ | 0.331 | 0.139 | 6.90 | 0.452 | 13.30 |
| Ours w/o lane-count prior | ✓ | ✓ | ✓ | ✗ | 0.009 | 0.031 | 3.94 | 0.115 | 5.10 |

**Count prior acts as a structural regularizer for masked completion.** Disabling the lane-count prior increases endpoint distance (0.115) and collisions (5.10) despite similar reconstruction losses, indicating that the prior primarily improves global structural calibration rather than local reconstruction. This matters for streaming outpainting because masked completion is invoked repeatedly; a small global count bias would accumulate into systematic density drift in long-horizon rollouts.

### A.15.3. EDGE-GATED RELATIONAL DiT: RELATION-AWARE ABLATIONS

**Relational logit bias targets lane connectivity failures.** Adding relational logit bias improves route length ($35.26 \to 37.10$), reduces endpoint distance ($0.301 \to 0.185$), and increases route validity ($87.36\% \to 92.10\%$). These improvements are consistent with the role of logit bias as a connectivity prior: it increases attention mass on edge-compatible pairs and reduces the chance of producing successor transitions with large geometric discontinuities.

**Edge-gated values suppress spurious message passing.** Adding the value gate further improves endpoint distance ($0.185 \to 0.142$) and reduces collision rate ($5.80\% \to 5.12\%$), while also improving multiple agent distributional JSD terms. Mechanistically, value gating limits the magnitude of feature aggregation along edge-inconsistent pairs, which prevents a small subset of incorrect edges from dominating latent updates. This is particularly important under large-step sampling, where each update must be reliable.

**Cross-type bias improves agent–lane alignment rather than only marginal realism.** Adding cross-type relational bias yields the largest improvement in the angular alignment statistic (Ang. JSD reaches its best value), which directly probes whether vehicle headings are consistent with nearby lanes. This supports that cross-type relational modeling is addressing a targeted geometric constraint (agent–lane alignment), not merely improving perceptual FD. The concurrent reduction in collision rate is expected because heading and lane alignment affect feasible packing and relative motion at initialization.

*Table 12.* **Ablation of relation-aware components in edge-gated relational DiT (Waymo).** We incrementally add relational modules to the factorized DiT backbone. **Cross-type bias** improves agent–lane geometric consistency (e.g., Ang. JSD) under the same map. Relational bias and value gating improve structural validity (e.g., route validity and endpoint distance) under large-step sampling. Our full model achieves the best structural integrity and safety.

| Variant | Relation-aware modules | | | | Perceptual | Map structure | | | Safety | Agent distributional JSD↓ | | | | |
|---|---|---|---|---|---|---|---|---|---|---|---|---|---|---|
| | Rel. bias | Value gate | Cross-type bias | Global fusion | FD ↓ | RouteLen. (m)↑ | Endpt. dist. (m)↓ | Route valid. (%)↑ | Coll. rate (%)↓ | Near. | Ang. | Len. | Wid. | Speed |
| Factorized DiT (Base) | ✗ | ✗ | ✗ | ✗ | 1.53 | 35.26 | 0.301 | 87.36 | 6.53 | 0.12 | 0.09 | 0.51 | 0.29 | 0.39 |
| + Relational logit bias | ✓ | ✗ | ✗ | ✗ | 1.35 | 37.10 | 0.185 | 92.10 | 5.80 | 0.11 | 0.09 | 0.42 | 0.25 | 0.35 |
| + Edge-gated values | ✓ | ✓ | ✗ | ✗ | 1.20 | 37.85 | 0.142 | 94.50 | 5.12 | 0.10 | 0.08 | 0.30 | 0.18 | 0.28 |
| + Cross-type rel. bias | ✓ | ✓ | ✓ | ✗ | 1.05 | 38.40 | 0.110 | 96.20 | 4.88 | 0.08 | **0.06** | 0.18 | 0.14 | 0.20 |
| + Global context (Ours) | ✓ | ✓ | ✓ | ✓ | **0.94** | **39.03** | **0.094** | **98.10** | **4.69** | **0.07** | **0.06** | **0.13** | **0.11** | **0.15** |

**Global fusion improves scene-level calibration.** Finally, global context fusion improves FD, route length, endpoint distance, and route validity simultaneously. This indicates that a lightweight global pathway helps calibrate scene-wide statistics that are otherwise difficult to infer from purely local attention, which is consistent with the reach/convenience topology objectives and with preventing density drift over repeated outpainting.

**Implication for streaming outpainting.** Because streaming repeatedly performs masked completion at the frontier, relation-aware attention reduces the probability of generating a single locally plausible but globally inconsistent edge transition, which would otherwise compound into topology breakage and failure to maintain long routes.

### A.15.4. LATENT GENERATIVE OBJECTIVE

*Table 13.* **Impact of the meanflow (Geng et al., 2025) objective on solver-free sampling.** We compare different training objectives under a fixed step budget. *Training Cost* is measured in GPU hours on 8×H20. Standard **Flow Matching (FM)** fails completely at 1-step (EPD 8.52m). **Distillation** is fast but blurs details (High Agent JSD). Our **MeanFlow (JVP+2-time)** incurs higher training cost (+50%) but enables the only viable 1-step generation (EPD 0.269m) that preserves agent statistics (JSD 0.136), enabling real-time streaming.

| Training recipe | Key design | | | | Train GPU-h ↓ | Steps at test | Waymo (same decoder) | | | |
|---|---|---|---|---|---|---|---|---|---|---|
| | MeanFlow $(u_\theta)$ | JVP $(\partial_t u)$ | Two-time $(t, \Delta)$ | Distill (1-step) | | | EPD (m)↓ | Agent JSD↓ | Time (ms)↓ | Valid (%)↑ |
| Flow matching | ✗ | ✗ | ✗ | ✗ | 320 | 1-step | 8.520 | 4.250 | **5.6** | 9.3 |
| | ✗ | ✗ | ✗ | ✗ | 320 | 12-step | 0.150 | 0.120 | 31.0 | **95.0** |
| MeanFlow (Identity) | ✓ | ✗ | ✗ | ✗ | **300** | 1-step | 0.554 | 0.499 | **5.6** | 63.2 |
| | ✓ | ✗ | ✗ | ✗ | 300 | 12-step | 0.120 | 0.110 | 31.0 | 96.5 |
| MeanFlow (Id + JVP) | ✓ | ✓ | ✗ | ✗ | 400 | 1-step | 0.293 | 0.312 | **5.6** | 68.0 |
| | ✓ | ✓ | ✗ | ✗ | 400 | 12-step | 0.125 | 0.131 | 31.0 | 84.2 |
| Shortcut (Frans et al., 2024) | ✗ | ✗ | ✗ | ✓ | 960 | 1-step | 0.350 | 0.469 | **5.6** | 47.3 |
| Ours (JVP + 2-time) | ✓ | ✓ | ✓ | ✗ | 480 | 1-step | 0.269 | 0.136 | **5.6** | 87.5 |
| | ✓ | ✓ | ✓ | ✗ | 480 | 12-step | **0.110** | **0.097** | 31.0 | 93.1 |

**MeanFlow identity improves one-step, but remains under-calibrated.** MeanFlow identity substantially improves the one-step regime (endpoint distance 0.554; validity 63.2%), indicating that interval-conditioned learning already reduces the extrapolation gap. However, the remaining quality gap suggests that predicting an average velocity alone is insufficient when the model must faithfully approximate boundary behavior at large step sizes.

**JVP correction targets boundary mismatch directly.** Adding the identity+JVP correction improves one-step quality $(0.554 \rightarrow 0.293)$ at the same inference time. This is consistent with the role of JVP: it provides a first-order correction that aligns the learned mean velocity with the boundary velocity required for a large-step update. Crucially, this reduces catastrophic structural errors without requiring additional solver steps at inference.

**Two-time conditioning is necessary for general large-step transport.** Our full recipe (JVP + two-time channels) further improves one-step quality and validity (endpoint distance 0.269; validity 87.5%) and also improves agent statistics. This indicates that explicitly conditioning on both the current time and the interval size is required to prevent step-size ambiguity: without the interval channel, the model must infer the effective step size implicitly from noise level alone, which is not identifiable under masking and mixed lane/agent latents.

**Training cost versus deployment value.** The full recipe increases training cost, but it is the only configuration that generates a usable one-step operating point while maintaining high validity. This trade-off is aligned with the streaming deployment constraint: training is amortized once, whereas inference is invoked at every outpainting trigger during rollouts.

**Step-budget sensitivity and deployment operating points.** Table 14 shows that meanflow provides a controllable quality–latency spectrum: one-step enables ultra-low latency streaming, while 3–5 steps recover much of the multi-step quality at a small additional cost. In contrast, DDPM reaches the best endpoint distance only at substantially higher latency, making it unsuitable for online outpainting. These results support our design choice to train for the large-step regime rather than relying on post-hoc acceleration of a diffusion-trained model.

*Table 14.* **Step-budget sensitivity of our latent generator (numeric companion).** Detailed metrics for Figure 5. While DDPM (100 steps) achieves the lowest EPD (Waymo: 0.080m; nuPlan: 0.072m), its latency (158ms) is the highest and limits online use. Our **MeanFlow-5** provides a favorable balance between fidelity and latency, while **MeanFlow-1** enables ultra-low latency streaming.

| | | Waymo | | | nuPlan | | |
|---|---|---|---|---|---|---|---|
| **Dynamics** | **Sampler / steps** | **EPD (m)↓** | **Agent JSD↓** | **Time (ms)↓** | **EPD (m)↓** | **Agent JSD↓** | **Time (ms)↓** |
| Flow (Rectified) | Flow-12 | 0.150 | 0.120 | 28 | 0.125 | 0.155 | 31 |
| | Flow-24 | 0.110 | 0.105 | 55 | 0.095 | 0.140 | 55 |
| | Flow-32 | 0.098 | 0.091 | 73 | 0.088 | 0.135 | 73 |
| | Flow-48 | 0.089 | **0.084** | 110 | 0.082 | **0.130** | 110 |
| MeanFlow | MeanFlow-1 | 0.269 | 0.136 | **5.6** | 0.224 | 0.171 | **6** |
| | MeanFlow-3 | 0.153 | 0.108 | 10 | 0.128 | 0.143 | 10 |
| | MeanFlow-5 | 0.130 | 0.102 | 14 | 0.109 | 0.137 | 14 |
| | MeanFlow-12 | 0.110 | 0.097 | 31 | 0.092 | 0.132 | 29 |
| DDPM | DDPM-100 | **0.080** | 0.092 | 158 | **0.072** | 0.185 | 158 |

*Table 15.* **Companion to Figure 5: FD / Fréchet Reach / Agent-JSD versus latency.** This table reports lane-graph fidelity (FD), topology-aware Fréchet Reach, and Agent-JSD mean (count excluded, following ScenDream-L) as a function of the solver-step budget on Waymo. Together with Figure 5 (which plots EPD / mean agent-JSD), it provides a direct view of the quality–latency trade-off. With a single MeanFlow step, VectorWorld is $\sim 28\times$ faster than ScenDream-L but trades off FD/Reach/JSD. Five-step MeanFlow nearly matches ScenDream-L on FD while being $\sim 7.5\times$ faster. The multi-step Flow operating point improves FD and Agent-JSD over ScenDream-L, while Fréchet Reach remains a topology gap (0.37 vs. 0.28).

| Method | FD ↓ | Fréchet Reach ↓ | Agent-JSD mean ↓ | Time (ms) ↓ |
|---|---|---|---|---|
| ScenDream-L (Rowe et al., 2025) | 1.38 | **0.28** | 0.205 | 160 |
| Ours (MeanFlow, 1 step) | 1.71 | 0.63 | 0.294 | **5.6** |
| Ours (MeanFlow, 3 steps) | 1.56 | 0.51 | 0.251 | 12.8 |
| Ours (MeanFlow, 5 steps) | 1.42 | 0.48 | 0.212 | 21.3 |
| Ours (Flow, multi-step) | **0.94** | 0.37 | **0.098** | 70.0 |

**Takeaways for deployment.** Table 14 provides the numeric companion to Figure 5 and is intended as an *operating-point guide* rather than a second copy of the objective analysis. Across datasets, increasing MeanFlow steps generates a smooth quality–latency trade-off: 1 step enables streaming ($\leq$ 30 ms/tile), 3–5 steps recover much of the multi-step fidelity at modest cost, and 12 steps approaches the best offline quality. In contrast, DDPM reaches its best endpoint distance only at substantially higher latency, making it unsuitable for online outpainting where generation is invoked repeatedly during rollout.

**Why one-step is non-trivial.** As discussed above (Table 13), objectives that only constrain instantaneous velocities are not explicitly trained for single large-step transport, which explains the failure mode of one-step flow matching.

### A.15.5. ΔSIM: FEASIBILITY–CONTROLLABILITY TRADE-OFFS

*Table 16.* **ΔSim component analysis on accuracy, controllability, and feasibility (Waymo).** We evaluate the impact of Hybrid Action Head (A+R) and Differentiable Kinematic Alignment Loss (DKAL). *Ctrl-Sim* baseline suffers from poor controllability ($\rho = -0.25$) and high latency. Our full **ΔSim** achieves the lowest ADE (1.72m) and highest controllability ($\rho = 0.53$). Crucially, **DKAL** significantly reduces closed-loop physical violations (e.g., Lat. Acc. 12.5% → 2.1%), ensuring stable long-horizon rollouts.

| Method | Design toggles | | | | Open-loop | | Closed-loop feasibility (%)↓ | | | | Efficiency | |
|---|---|---|---|---|---|---|---|---|---|---|---|---|
| | RTG (1-pass) | Hybrid A+R | Logit shaping | DKAL loss | ADE (m)↓ | $\rho\uparrow$ | Coll. | Yaw rate | Lat. acc. | Lon. jerk | Lat. (ms)↓ | Fwd ↓ |
| Ctrl-Sim (Rowe et al., 2024) | ✗ | ✗ | ✗ | ✗ | 2.80 | -0.25 | 15.2 | 8.5 | 14.2 | 10.5 | 36.9 | 2.0 |
| ΔSim (discrete-only) | ✓ | ✗ | ✓ | ✓ | 2.62 | 0.39 | 8.5 | 4.2 | 5.5 | 6.2 | **21.7** | **1.0** |
| ΔSim (hybrid A+R, no DKAL) | ✓ | ✓ | ✓ | ✗ | 2.10 | 0.24 | 10.1 | 6.8 | 12.5 | 8.9 | 26.1 | **1.0** |
| ΔSim (hybrid A+R, no shaping) | ✓ | ✓ | ✗ | ✓ | 1.95 | 0.45 | 5.2 | 2.5 | 3.1 | 4.5 | 26.1 | **1.0** |
| **ΔSim (Ours)** | ✓ | ✓ | ✓ | ✓ | **1.72** | **0.53** | **2.9** | **1.2** | **2.1** | **3.2** | 26.1 | **1.0** |

**What this ablation isolates.** Table 16 decomposes the behavior model into four mechanisms: single-pass return conditioning, hybrid discrete–continuous actions, inference-time logit shaping, and differentiable kinematic alignment. We report three axes that matter specifically for long-horizon closed-loop simulation: trajectory accuracy (ADE), controllability ($\rho$), and feasibility under rollout (violation rates; definitions in Section A.12).

**Single-pass return conditioning improves efficiency without sacrificing control.** The baseline requires two forward passes and has poor controllability ($\rho = -0.25$). Our single-pass return conditioning reduces forward passes to one and improves controllability ($\rho = 0.39$ in the discrete-only variant), indicating that the control signal is injected into the same computation that produces actions, rather than being used as an auxiliary prediction that is later re-fed.

**Hybrid action heads remove the vocabulary-resolution bottleneck.** The discrete-only variant has higher ADE (2.62), consistent with quantization error from a finite k-disks vocabulary. Introducing the residual refinement head reduces ADE ($2.62 \rightarrow 2.10$ and below), confirming that continuous refinement improves precision without requiring a larger discrete vocabulary, which would increase both memory and sampling entropy.

**Why hybrid actions require physics-aligned training.** A hybrid head increases expressivity; without a feasibility-aligned learning signal, the model can allocate probability mass to precise but physically implausible transitions, such as high lateral acceleration, that are weakly penalized by pure imitation losses. This is visible in the hybrid-no-DKAL row, where feasibility violations increase sharply (e.g., lateral acceleration 12.5%) even though ADE improves. This pattern indicates that expressivity alone is insufficient; it must be paired with an inductive bias that matches the closed-loop feasibility objective.

**DKAL provides train–infer alignment for feasibility.** Adding DKAL substantially reduces feasibility violations and improves overall stability of rollouts. Notably, DKAL reduces violations even when logit shaping is disabled, indicating that the model learns to rank actions in a feasibility-consistent way, rather than relying on inference-only heuristics. When both DKAL and shaping are enabled, the model achieves the best combined operating point: lowest ADE, highest controllability, and lowest violation rates.

**Implication for long-horizon stability.** Feasibility violations are a compounding-error mechanism: small rates per step become dominant over kilometer-scale horizons. The reductions in yaw-rate and lateral-acceleration violations therefore directly explain the improved stability observed in the long-horizon closed-loop results.

*Table 17.* **Policy training value with cross-backend evaluation (Waymo).** To reduce the risk of simulator overfitting, we evaluate policies across different backends under matched horizons. A policy retrained in VectorWorld improves success not only in VectorWorld but also transfers to log replay and ScenDream backends (see Panel A). Panel B further evaluates long-horizon performance (1 km+) in VectorWorld.

*Panel A: Fixed-horizon cross-backend evaluation (200 m).*

| Policy | Train backend | Train data (#eps) | Train GPU-h | Eval: Log Replay C/O/Succ (%) | Time (h) | Eval: ScenDream-L (neg.) C/O/Succ (%) | Time (h) | Eval: VectorWorld (neg.) C/O/Succ (%) | Time (h) |
|---|---|---|---|---|---|---|---|---|---|
| PPO | Waymo Log | 100k | **64** | 29.3 / 6.9 / 63.8 | **0.4** | 59.0 / 9.0 / 32.1 | 5.8 | 65.5 / 9.5 / 25.0 | 1.2 |
| PPO | ScenDream-L | 100k | 480 | 19.3 / **5.9** / 74.8 | **0.4** | 45.2 / 8.5 / 46.3 | 5.8 | 40.9 / 7.6 / 48.5 | 1.2 |
| PPO | **VectorWorld** | 100k | 160 | **18.1** / 6.2 / **75.7** | **0.4** | **38.5 / 5.6 / 56.3** | 5.8 | **27.1 / 5.0 / 66.9** | 1.2 |

*Panel B: Long-horizon evaluation (1 km+; VectorWorld only).*

| Policy (Eval: VectorWorld neg.) | Eval time (h) | Coll. (%)↓ | Offroad (%)↓ | Succ. (%)↑ |
|---|---|---|---|---|
| Pretrained PPO (Baseline) | 1.0 | 65.1 | 9.2 | 25.7 |
| PPO retrained in VectorWorld (Ours) | 1.0 | **35.1** | **7.9** | **56.0** |

*C/O/Succ*: Collision Rate / Offroad Rate / Success Rate.
*Time*: Wall-clock hours for evaluating 1000 episodes. Note VectorWorld's efficiency (1.2h) vs ScenDream (5.8h).

*Table 18.* **Failure reproduction across backends.** We verify if failures discovered in simulation correspond to realistic crash types. We match generated failure scenarios to the nearest log scenarios based on map and agent topology. VectorWorld failures show high **Type Agreement** (78.5%) with log data, indicating that our simulator exposes realistic safety-critical events rather than simulation artifacts.

| Source of failures | Matching and reproduction diagnostics | | | | Outcome agreement | |
|---|---|---|---|---|---|---|
| | #Fail episodes | Log match (%) | Init. state dist. (L2) ↓ | Route match (%) | Fail repro. (%) | Type agreement (%) |
| Log replay (reference) | 100 | 100.0 | 0.00 | 100.0 | 100.0 | 100.0 |
| ScenDream-L (neg. tilt) | 528 | 65.2 | 1.42 | 82.5 | 45.2 | 52.1 |
| VectorWorld (neg. tilt) | 655 | **88.5** | **0.92** | **95.4** | **72.0** | **78.5** |

### A.15.6. CLOSED-LOOP VALIDATION BEYOND SELF-CONSISTENCY

**Cross-backend evaluation addresses self-consistency.** A common threat to validity for learned simulators is self-consistency: a policy might appear strong only because it exploits simulator-specific artifacts. Table 17 evaluates the same policies across multiple backends under matched horizons. A policy retrained in VectorWorld improves success not only in VectorWorld but also transfers to log replay and ScenDream backends, indicating that the learned improvements are not tied to a single simulator implementation.

**Why transfer is expected in our design.** VectorWorld reduces two major sources of simulator-specific bias: (i) initialization mismatch, by providing motion-history warm starts consistent with the behavior-model context, and (ii) structural drift, by enforcing relational consistency in generation and feasibility alignment in behavior. These design choices reduce the incentive for policies to overfit to artifacts, such as unrealistic transient dynamics or topology breakage, which in turn improves transfer.

**Long-horizon evaluation is a differentiator, not a cosmetic extension.** Panel B of Table 17 shows that the retrained policy maintains substantially higher success over a 1 km-scale horizon. This regime is precisely where compounding errors become dominant and where a simulator must be evaluated as a system rather than as a static generator.

**Failure realism diagnostics.** Table 18 provides a failure-reproduction diagnostic: we match simulator-discovered failures to the nearest logged scenarios under map and agent-topology similarity, and measure agreement in failure type. VectorWorld achieves higher matching rates and higher type agreement than the baseline generator, indicating that the discovered failures are structurally similar to real-world events rather than being dominated by simulator-specific artifacts. This supports using

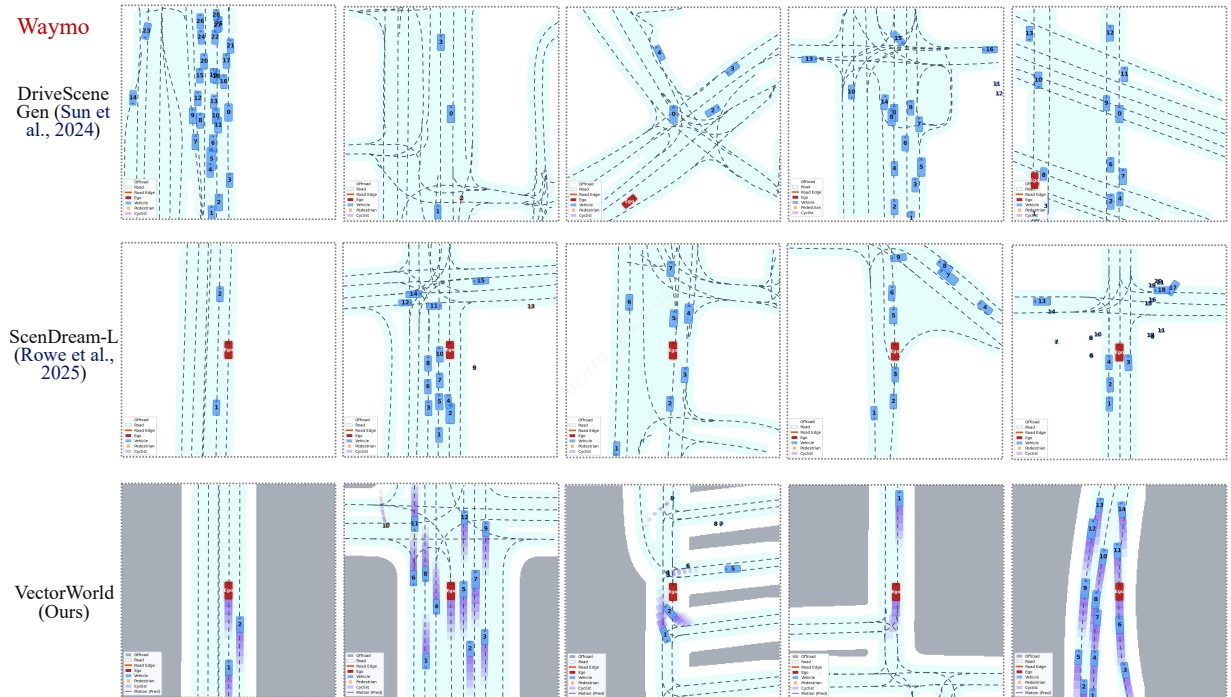

*Figure 8.* **Waymo qualitative comparison for initial scene generation** (64 m × 64 m). We compare VectorWorld against representative vectorized scene generators and the Waymo reference. The visualization highlights lane connectivity at intersections, agent–lane geometric consistency, and the plausibility of the initialized interaction state, including short motion history when available.

VectorWorld as a stress-test backend for policy improvement rather than only as a diversity tool.

### A.16. Qualitative Results and Visual Analysis

This section provides qualitative visualizations that complement the quantitative metrics in Tables 1, 2 and 10. We organize the analysis around the same deployment constraints that motivate VectorWorld: (i) topology-consistent initialization, (ii) controllability under fixed map conditions, (iii) real-time end-to-end latency for online generation, and (iv) long-horizon stability under repeated streaming outpainting.

**Rendering protocol (common to all figures).** Unless otherwise stated, all panels are rendered from the decoded vector-graph representation: lanes are shown as centerline polylines, agents are shown as oriented boxes, and, when available, short motion histories are shown as agent-frame polylines re-projected to the ego frame at initialization. Importantly, the visualizations do not rely on manual editing; they reflect the decoded simulator state after the same deterministic validity filters described in Section A.10. This makes the qualitative observations directly comparable to the offline validity and topology metrics.

#### A.16.1. WAYMO: INITIAL-SCENE GENERATION COMPARISONS

**Topology consistency at intersections is an edge-level property.** The Waymo panels in Figure 8 illustrate a recurrent failure mode of vector-graph generation: local lane geometry can appear plausible, yet successor transitions become discontinuous or inconsistent at intersections and merges. This is precisely an edge-defined constraint: connectivity is encoded by $\mathcal{E}_{\text{L2L}}$, not by marginal lane-node appearance. VectorWorld reduces such discontinuities by explicitly modulating attention with typed edge attributes (Section 3.3 and Equation (2)), which is consistent with the improvements in endpoint distance and route validity in Tables 1 and 10. Qualitatively, this generates lane graphs that remain traversable across junctions without relying on heuristic stitching.

**Agent–lane alignment and initialization feasibility co-vary in closed loop.** The same figure also reveals that agent placement and orientation errors are not isolated artifacts: misalignment to nearby lane tangents (e.g., heading inconsistent

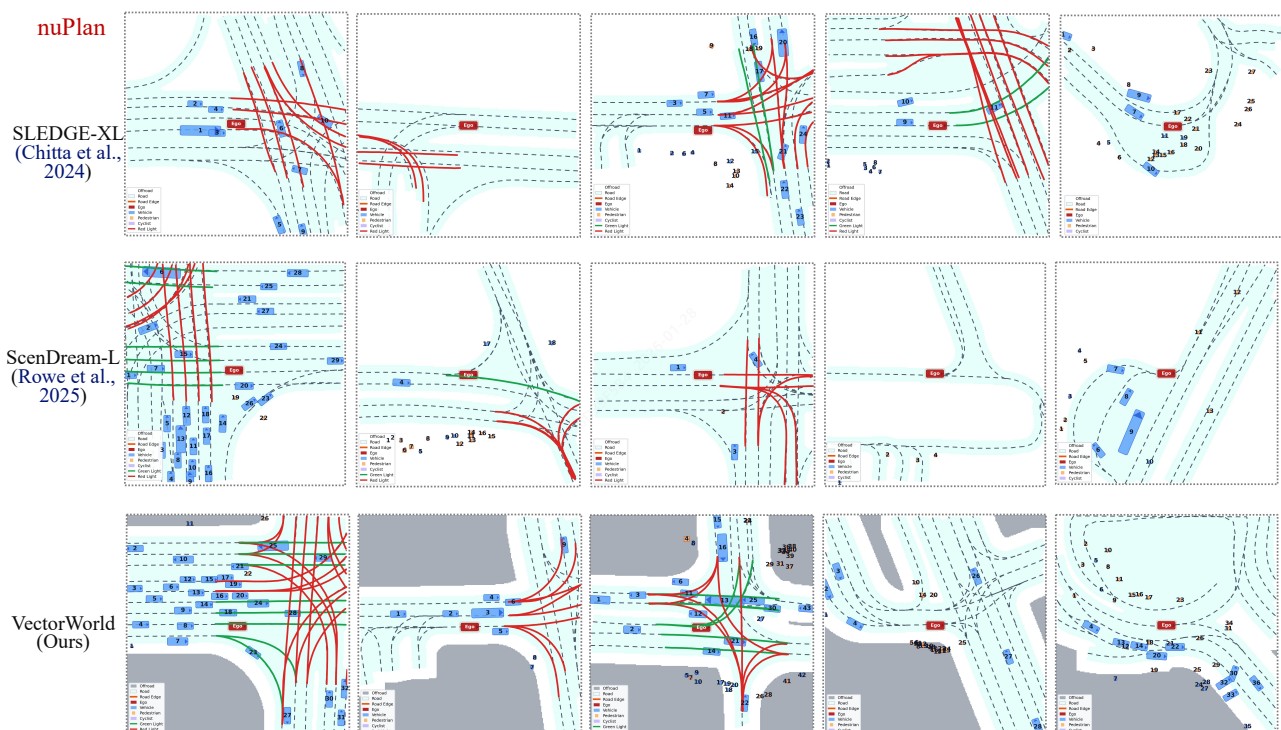

*Figure 9.* **nuPlan qualitative comparison with traffic-light semantics.** Lane centerlines are colored by traffic-light state (green vs. red) to visualize control-relevant map semantics. We compare VectorWorld with strong baselines under the same vectorized rendering.

with centerline direction) increases the probability of overlaps and early collisions after the first few simulator steps. VectorWorld's generator benefits from cross-type relational modeling, which conditions agent tokens on lane context and vice versa, making agent orientation and lane assignment more consistent at initialization. This qualitative pattern matches the lower initialization collision rate in Table 1 and the improved angular/size/speed distribution alignment in Table 10 (Waymo).

**Why interaction-state initialization is visually diagnostic (beyond FD).** A key difference between VectorWorld and snapshot-only generators is that VectorWorld instantiates each agent with a compact motion-history code (Section 3.2). While FD primarily measures distribution alignment in an embedding space, the interaction-state interface determines whether the downstream behavior model receives in-distribution temporal context at $t = 0$. In Figure 8, VectorWorld's initialized agents exhibit consistent short-horizon motion cues, which reduces the "zero-history" prior implicitly assumed by history-conditioned policies. This mechanism is causally linked to the reduced jerk and improved closed-loop success when WarmStart is enabled (Table 2, Section I).

### A.16.2. NUPLAN: TRAFFIC-LIGHT SEMANTICS AND LANE−AGENT CONSISTENCY

**Traffic-light semantics create structured conditional dependencies.** Traffic signals in nuPlan induce a constraint that couples distant elements: the lane state (red/green) constrains which lane segments are currently traversable, which in turn constrains feasible agent intent and placement (e.g., queue formation at red lights vs. flowing traffic on green). This dependency is not captured by purely local appearance matching, and is particularly brittle under repeated generation because a single inconsistent lane state can propagate into unrealistic agent motion in closed loop.

**VectorWorld preserves signal-conditioned lane structure and its downstream implications.** In Figure 9, VectorWorld produces lane centerlines whose traffic-light coloring is consistent across the intersection structure, and the resulting agent configurations remain compatible with the implied right-of-way constraints. Mechanistically, this is expected from two design choices: (i) the heterogeneous graph representation makes lane semantics explicit at the node/edge level (Section 3.1), and (ii) edge-gated relational attention provides a controlled pathway for propagating lane semantics into cross-type

Agent Controllability (Same Map As Condition)

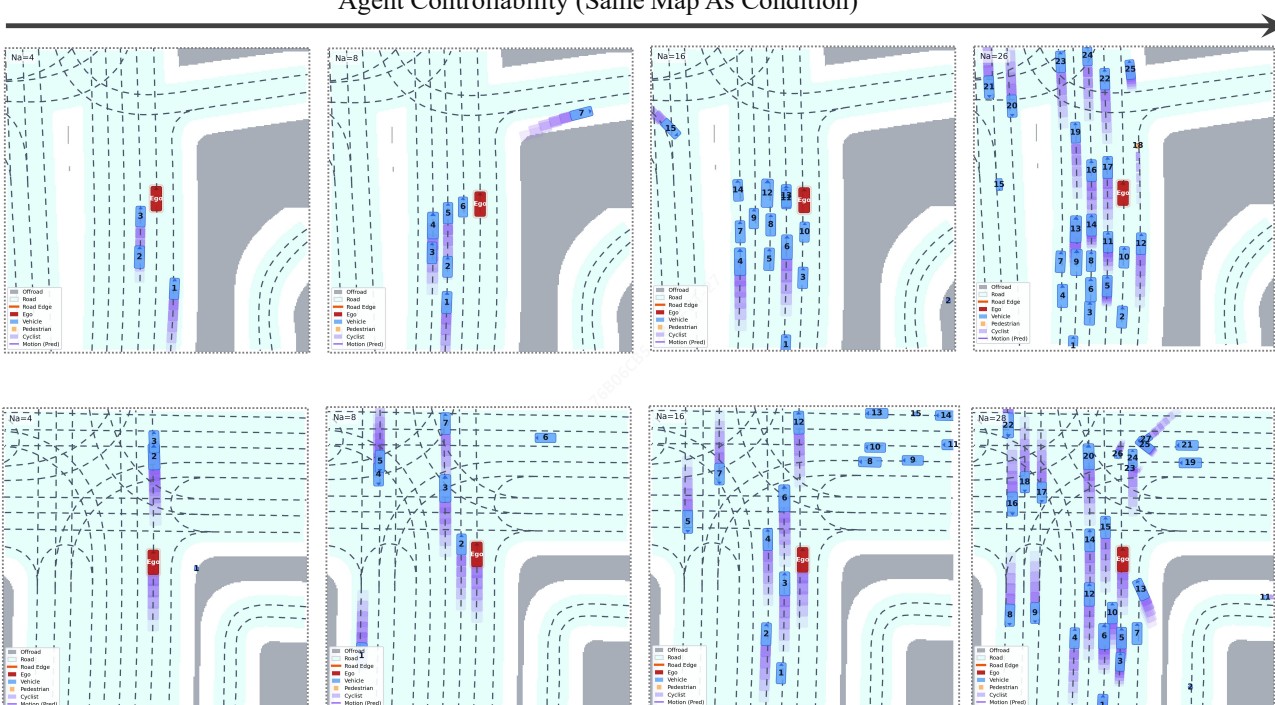

*Figure 10.* **Agent controllability under fixed map conditioning.** The map (lane graph) is held fixed as a condition, while the agent configuration is generated under different controllability settings. This visualization isolates controllability from map diversity by using the same conditioned lanes across all panels.

interactions (Section 3.3). Qualitatively, this generates fewer cases where agents are initialized with headings or lane assignments that contradict the current signal state.

**Why this matters for closed-loop evaluation.** Signal-conditioned inconsistencies can appear benign in single-frame offline metrics, yet they directly affect closed-loop behavior: a history-conditioned planner reacts strongly to right-of-way cues, and inconsistent initialization can trigger abrupt braking or unsafe merges. The traffic-light visualization therefore serves as a targeted diagnostic for "control-relevant realism," complementing FD/topology metrics. It also explains why we emphasize structure and safety metrics alongside perceptual alignment in Table 1.

### A.16.3. MAP-CONDITIONED CONTROLLABILITY OF AGENT GENERATION

**Controllability should be evaluated under a fixed map to avoid confounding.** A common confound in scenario generation is that apparent "control" can be explained by changes in the generated map itself (e.g., different lane topology induces different feasible behaviors). Figure 10 fixes the lane graph as a condition and varies only the controllability input that affects agent generation. This isolates whether the model can modulate agent intent and configuration while maintaining compatibility with the same geometric constraints.

**Why VectorWorld supports map-conditioned control.** VectorWorld's interface separates (i) conditioned structure (clamped lane tokens) from (ii) sampled content (agent tokens), and uses a unified masked-completion mechanism in latent space (Sections 3.1 and 3.5). As a result, the generator can keep lane topology invariant while sampling alternative agent sets that remain aligned with lane geometry. The motion-aware gated VAE further stabilizes the controllability effect by representing agent initialization as an interaction state: changing the control input modifies not only the instantaneous pose distribution, but also the implied short-horizon momentum at $t = 0$, which is the quantity that reactive NPC policies and ego planners are most sensitive to.

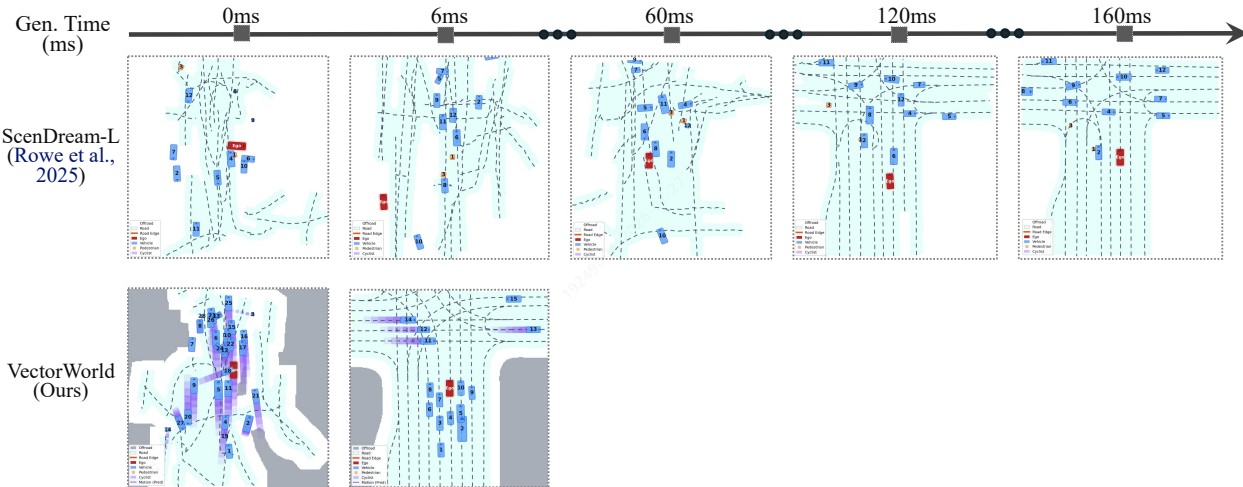

*Figure 11.* **End-to-end generation latency (timeline view).** Qualitative snapshots under increasing time budgets for a $64\,\text{m} \times 64\,\text{m}$ tile. VectorWorld reaches the final lane–agent vector scene with one solver-free MeanFlow step plus one VAE decode ($\approx 6\,\text{ms}$), whereas multi-step diffusion baselines only converge at substantially higher budgets (e.g., $\sim 160\,\text{ms}$).

**Feasibility is a necessary complement to controllability.** Controllability is only useful if the resulting agent configurations remain feasible under rollout. In VectorWorld, feasibility is enforced at two levels: (i) the generator improves initialization validity (lower static collision rate), and (ii) $\Delta$Sim suppresses kinematically infeasible actions over long horizons via physics-aligned learning and logit shaping (Section 3.6 and Equations (9) and (10)). This is consistent with the fact that control sweeps in Table 2 can systematically modulate failure rates without requiring shorter rollouts or early termination due to simulator instability.

A.16.4. END-TO-END LATENCY COMPARISONS ($64\,\text{m} \times 64\,\text{m}$ SCENES)

**Measurement protocol.** We report wall-clock latency per $64\,\text{m} \times 64\,\text{m}$ candidate scene with GPU synchronization. The timing includes (i) latent sampling and (ii) exactly one VAE decode into the vector-graph representation; it excludes data loading and any optional deterministic filtering so the reported numbers isolate model inference cost.

**Key result.** As shown in Figures 11 and 12, VectorWorld operates in the single-digit millisecond regime: one MeanFlow update at $(t, \Delta) = (1, 1)$ followed by one decode yields a complete lane–agent tile in $\approx 6\,\text{ms}$ per candidate.

**Why the speedup is structural (not an implementation artifact).** Our generator is explicitly trained for solver-free large-step transport and deployed with the one-step update $z_0 = z_1 - u_\theta^{\text{cfg}}(z_1; t{=}1, \Delta{=}1, c)$ (Equation (6)), so inference cost is essentially one DiT forward pass plus one decode. In contrast, diffusion-style baselines must iterate many denoising steps, each invoking a comparable backbone; therefore, latency scales roughly linearly with the solver step budget even when the decoder and output representation are matched. This end-to-end latency is the binding constraint in streaming outpainting, where generation is invoked repeatedly during closed-loop rollout.

A.16.5. REAL-TIME STREAMING OUTPAINTING CASE STUDIES ($1\,\text{km}+$)

**Challenging scenarios mined from VectorWorld generation.** Figure 13 shows representative scenarios selected from VectorWorld-generated environments. These cases feature dense interactions, complex intersection topology, and tight spatiotemporal gaps that empirically stress a fixed PPO ego planner. They complement the system-level statistics in Table 2 by visualizing the kinds of planner-relevant edge cases enabled by streaming outpainting.

**Streaming exposes failure modes that short-horizon generation cannot reveal.** A fixed short-horizon generated environment can look plausible even if small structural errors exist. Streaming outpainting amplifies these errors because the system repeatedly conditions on previously generated content. This makes seam consistency, topology preservation, and long-horizon feasibility the dominant factors, rather than single-frame perceptual alignment.

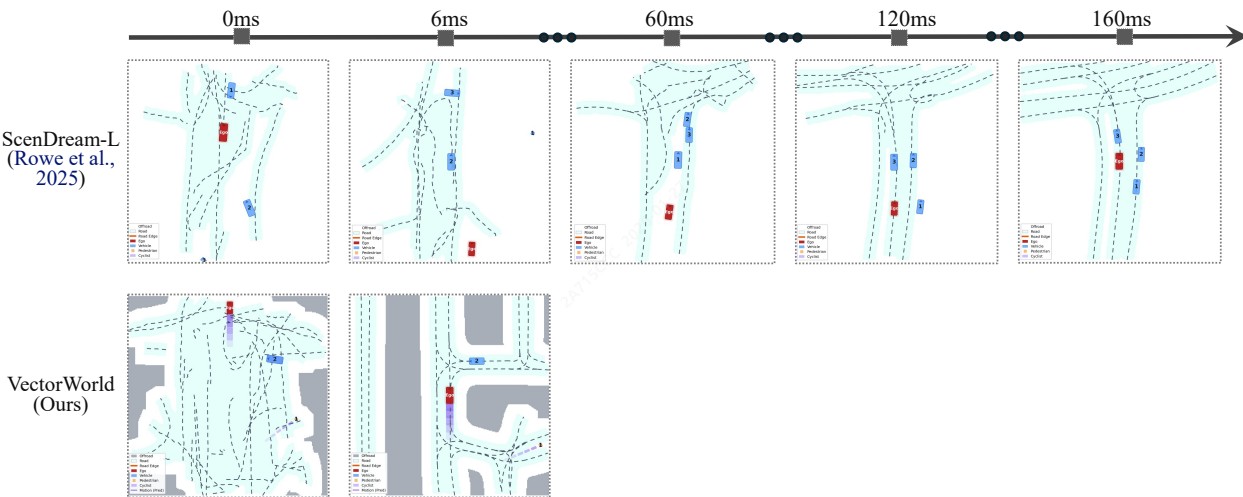

*Figure 12*. **End-to-end generation latency.** A different scene instance rendered with the same timing protocol (latent sampling + one VAE decode). The result is consistent: VectorWorld completes a full $64\,\mathrm{m} \times 64\,\mathrm{m}$ lane–agent tile in $\approx 6\,\mathrm{ms}$ via a single MeanFlow update, while solver-based diffusion requires many iterative denoising calls.

**Why clamping + constant-path conditioning reduces structural drift.** VectorWorld's outpainting clamps previously observed latents and enforces a constant probability path for conditioned tokens (Section 3.4 and Equation (3) and the masked-token construction). This mechanism prevents conditioned structures from drifting over repeated updates. In Figures 14 and 15, the lane graph remains continuous across tile boundaries, which is consistent with the strong endpoint-distance and route-length metrics reported in the main paper.

**Dynamic agent management and intent control require an interaction-state interface.** In long-horizon rollouts, the active agent set must change: agents behind the ego should be removed, and new agents should appear ahead. However, naive insertion based on static states tends to introduce cold-start artifacts, such as non-physical jerks, that destabilize interactions. VectorWorld mitigates this by generating agents as interaction states with motion-history codes (Section 3.2), so newly inserted agents can be warm-started with implicit momentum consistent with the behavior model's context window. This is qualitatively visible as stable agent motion around insertion events, and quantitatively reflected by reduced jerk with WarmStart (Table 2).

**Why long-horizon stability depends on feasibility-aligned behavior modeling.** Even if maps are outpainted consistently, long-horizon stability is ultimately limited by the compounding rate of kinematic violations in NPC actions. $\Delta$Sim reduces these compounding errors via hybrid action tokenization and feasibility-aligned learning (Section 3.6). The streaming case studies therefore serve as a system-level validation: they demonstrate that the generator ($\sim$ms latency), the interface (warm-started interaction states), and the behavior model ($\Delta$Sim) jointly satisfy the constraints required for kilometer-scale closed-loop simulation.

### A.17. Limitations and Safety Considerations

**Representation coverage.** VectorWorld currently operates on a compact centerline-based map representation and does not model all map elements (e.g., detailed road boundaries, curb geometry, and some fine-grained traffic control semantics). As a result, certain planner-relevant constraints may be under-specified in the generated environment, especially in dense urban layouts.

**One-step streaming trade-offs.** Our design prioritizes solver-free large-step sampling for streaming outpainting. While this enables real-time deployment, it can leave residual gaps in fine-grained distribution matching. For example, VectorWorld does not minimize every marginal agent JSD term on nuPlan (e.g., Near./Lat. in Table 10), and the Conve. topology discrepancy can remain higher than some baselines. These gaps motivate improving fine-grained alignment under the same streaming constraint, rather than relaxing latency via multi-step solvers.

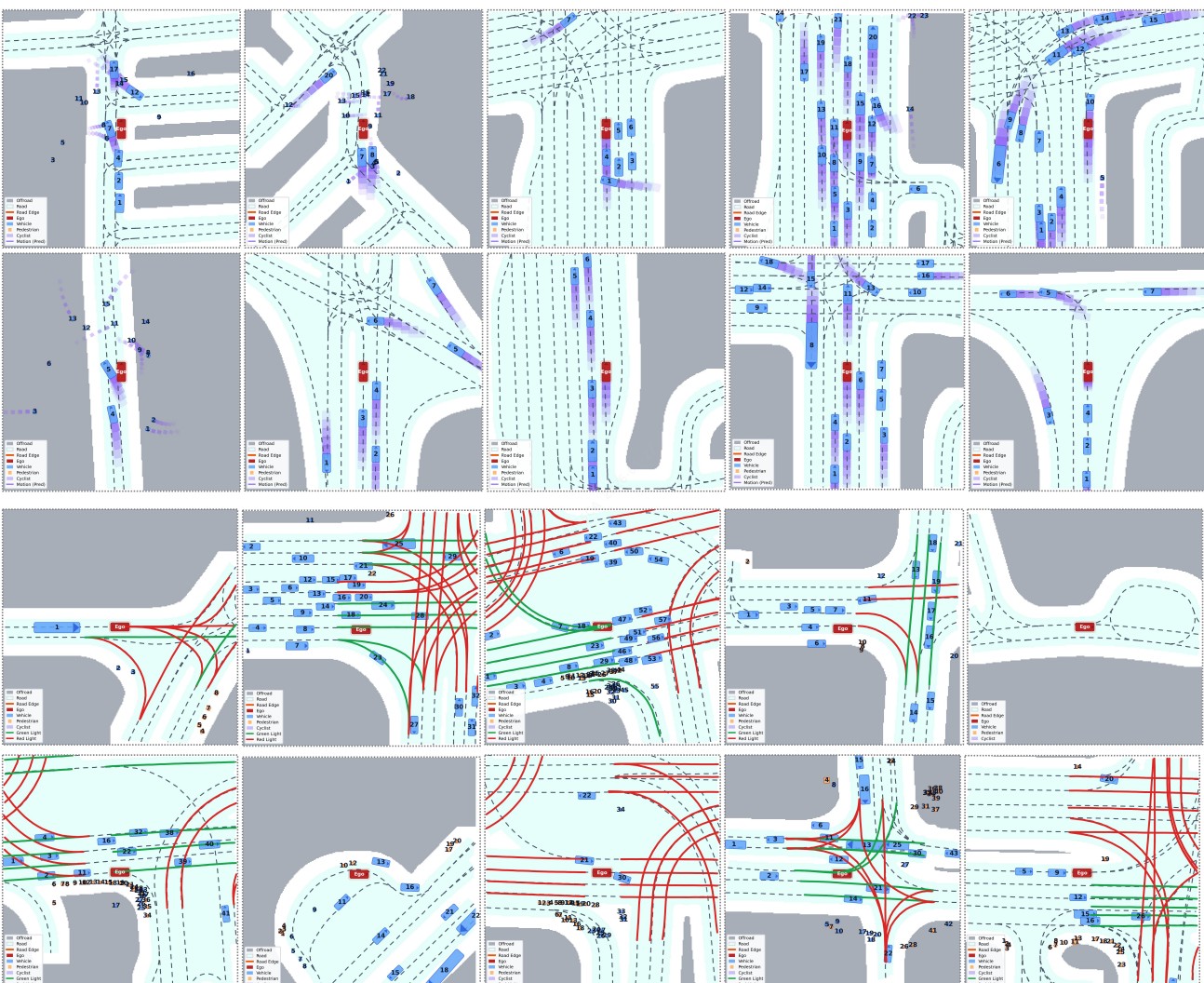

*Figure 13.* **Planner-relevant challenging scenarios sampled from VectorWorld.** We render a subset of VectorWorld-generated scenes using the protocol in Section A.16. These scenarios illustrate diverse intersection geometries, dense multi-agent interactions, and tight gaps that are valuable for stress testing and training an ego PPO planner in closed loop.

**Behavior-model fidelity under extreme maneuvers.** ΔSim uses delta-based updates and feasibility-aligned shaping to reduce kinematic violations, but it does not guarantee strict physical feasibility under all extreme conditions (e.g., rare high-speed evasive maneuvers). Closed-loop results should therefore be interpreted as the fidelity of the proposed generative-simulation pipeline under the stated assumptions and thresholds.

**Safety considerations and intended use.** VectorWorld is designed for research on simulation-based evaluation and training, not for direct deployment in real vehicles. Generated scenarios may contain rare, safety-critical interactions; this is useful for stress testing but also creates a misuse risk if employed to design adversarial traffic behaviors outside a controlled research context. We recommend (i) using VectorWorld-generated scenarios only within sandboxed simulation, (ii) reporting configured feasibility thresholds and filtering rules, and (iii) validating any learned policy improvements with cross-backend or log-replay evaluation (e.g., Appendix A.15.6) to reduce simulator-specific overfitting.

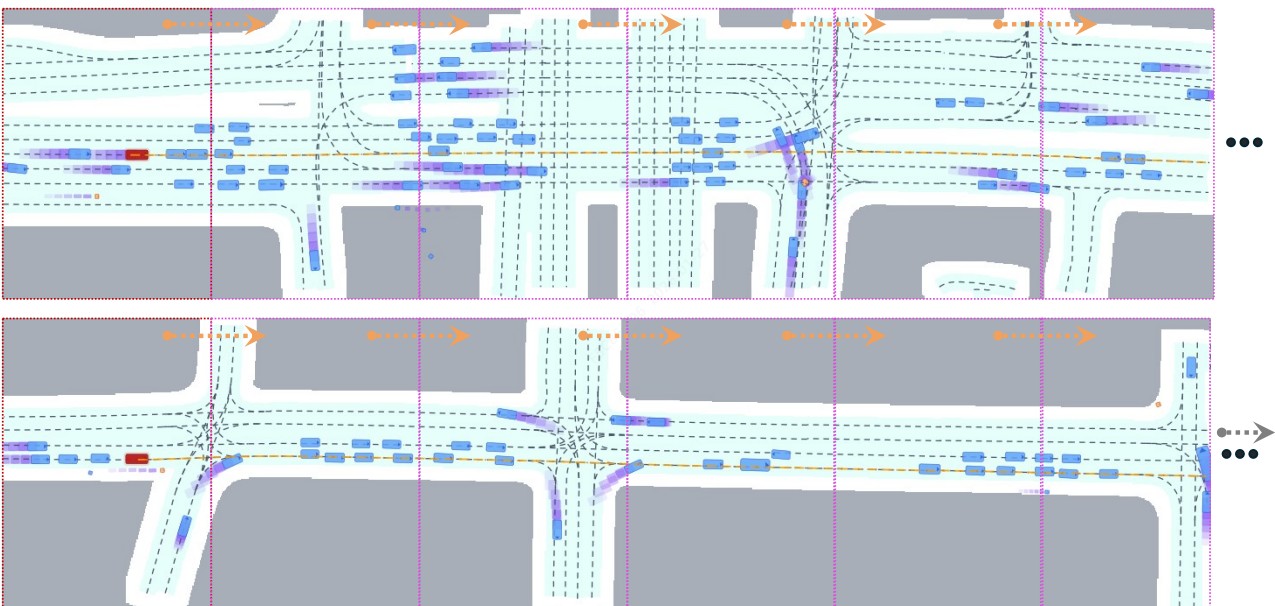

*Figure 14.* **Streaming outpainting in closed-loop rollout (case study 1).** The simulator repeatedly outpaints new lane–agent tiles near the frontier while clamping the observed region. The visualization highlights seam consistency and stable long-horizon progression.

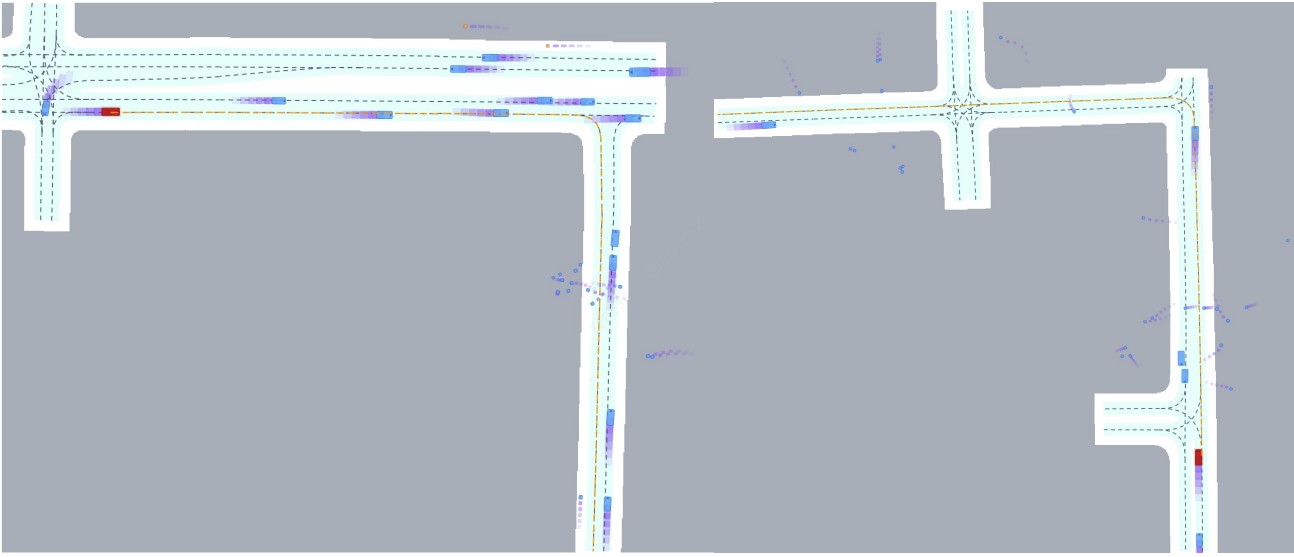

*Figure 15.* **Streaming outpainting with dynamic agent set and per-agent intent control (case study 2).** The visualization illustrates stable long-horizon expansion with agent insertion/removal and controllable behaviors, enabled by the interaction-state interface.

