# OpenReview forum: "VectorWorld: Efficient Streaming World Model via Diffusion Flow on Vector Graphs"
_ICML.cc/2026/Conference — ICML 2026 spotlight_

### Official Review · Reviewer_qTsg · 2026-03-04

**Soundness:** 3
**Presentation:** 3
**Significance:** 3
**Originality:** 3
**Overall Recommendation:** 4
**Confidence:** 3

**Summary:**

It is well known in autonomous vehicles research that accurate realistic simulation is a key bottleneck to current methods. Specifically, we want to be able to believably simulate what the traffic scene looks like from a given spatio-temporal location which is off the historic trajectory in our collected training dataset. This is where generative models for traffic evolution come in. They hold the promise to be able to generate realistic and plausible dynamic continuations of the evolution of the current scene. VectorWorld is such a model - it generates on demand the future vector-graph elements corresponding to the lanes and agents around the ego-vehicle in the scene.

Its technical design is centered around solving 3 challenges:
- **History-free initialization**: the restriction to not use historical motion information. This is handled by augmenting each agent with a latent learned code, obtained from a motion-aware VAE.
- **Latency**: the necessity to generate the scene quickly, in an online manner. This is handled by a specialized DiT, trained with MeanFlow and JVP-based large-step objectve.
- **Long-horizon feasibility**: the small errors that accumulate into big errors over longer generation. This is handled by a NPC-policy called Delta-Sim.

Experiments are large-scale, on the NuPlan and Waymo Open Motion datasets.

**Compliance With Llm Reviewing Policy:**

Affirmed.

**Key Questions For Authors:**

1. Could you explain more concretely how the history-free initialization creates the "jerk" spikes you mentioned? I don't fully understand why this happens.
2. What are EPD and JSD? I believe these are not explained.
3. When generating the future, do you generate only lanes and road elements or also agents? Do you generate new agents showing up into the scene?
4. In eqn. 10 what does the cost C represent? How is it computed? Why do you need to center the logits and the cost?

**Limitations:**

yes

**Strengths And Weaknesses:**

Strengths
===
1. The method is well-designed, with careful attention to detail and a clear understanding of the problems that need to be solved, and how each component addresses them.
2. The experimental evaluation is strong. The NuPlan and Waymo Open Motion datasets are core large-scale benchmarks and it is good to see that the authors obtain good results on them. Also, I appreciate the selection of the baselines (SLEDGE, ScenDream). These are strong baselines and it is reassuring that the authors manage to improve over them.
3. The paper clearly identifies deployment constraints that are often overlooked in generative world model research (initialization mismatch, real-time latency, and long-horizon stability).

Weaknesses
===
1. The method itself is, perhaps, more complicated than it needs to be. I did not fully understand whether the complexity is justified. For example, consider the components: agent encoder, lane encoder, factorized scene transformer, Gaussian latent heads in the VAE, edge-gated relational DiT with relational attention, return-to-go distribution, JVP-based large-step objective, FiLM modulator, hybrid action heads, differentiable kinematic cost, ... I understand that all of these are well-motivated individually, but as presented the method seems to be overly engineered a bit. Can we see simple ablations showing how each component contributes to the score gains?
2. In terms of writing style I think the paper comes off as too technical. I think that a paper needs to contain information on three levels: 1) high-level information such as motivation, purpose, inputs-outputs; 2) Mid-level information such as algorithms and component design; 3) Low-level information, technical details useful only to those who will implement the algorithm from the paper. I believe the current writing focuses too much on 2) and 3), and not enough on 1). The language is often overcomplicated and simplifying it will improve the communication with researchers outside of the autonomous driving setting.

Note: overall I think it's a very good paper, well-motivated with good experimental results and plenty of technical contributions. The weakness I mentioned are rather tied to my personal preference. I do not have objections with the actual technical design, which I actually find clever and well-motivated.

---

> ### Author Rebuttal · Authors · 2026-03-26
>
> Thank you for the helpful comments. We will revise to make the high-level story and definitions clearer. Point-by-point responses:
>
> **Q: Clarifying the main system contribution and the role of the major modules**
> **A:** The main contribution is **one unified system**: a real-time vectorized closed-loop simulator. The major modules are the three interfaces required to make it work, not unrelated add-ons.
>
> | Major module | Representative gain | System role |
> |---|---|---|
> | Interaction-state initializer | Init. coll. $9.45 \to 4.69$; jerk $16.6 \to 9.6$ | History-compatible start for history-conditioned policies |
> | Relation-aware one-step generator | $5.6$ ms/tile; valid scenes $9.3\% \to 87.5\%$; route $87.36 \to 98.10$ | Real-time structured frontier outpainting |
> | Physics-aligned $\Delta$Sim | ADE $2.80 \to 1.72$; lat.-acc. viol. $14.2\% \to 2.1\%$ | Stable kilometer-scale NPC rollout |
>
> We will revise the introduction to make this **one-system / three-module** structure explicit and separate system-level ideas from implementation details.
>
> **Q: Enriching high-level context while maintaining technical rigor**
> **A:** Thank you. We will foreground motivation, purpose, and I/O more clearly. Log replay cannot support interactive evaluation beyond recorded data; existing generative simulators are too slow or unstable for closed-loop use. VectorWorld bridges this gap: input is the ego-centered vector scene (lanes + agents), motion history, and ego behavior; output is reactive NPC motions and dynamically generated map–agent tiles beyond the log horizon. We will simplify component design in the main text and move details to the appendix.
>
> **Q: Why does history-free initialization create jerk spikes?**
> **A:** Both the ego planner and $\Delta$Sim are **history-conditioned**: they read a short motion window to infer velocity, curvature, braking intent, etc. At $k=0$ the simulator must provide a full window since these models expect **fixed-length history**. With history-free init, missing frames are fabricated (zero-padding or repetition). For a moving agent this creates an OOD input: the current state says the agent is moving while the pseudo-history implies it was static or moving differently. The first action thus becomes a large corrective step (sudden braking/acceleration/steering), producing a **jerk spike** ($j_t = (a_t - a_{t-1})/\Delta t$). Our motion code reconstructs a dynamically consistent pseudo-history, reducing jerk from **16.6 to 9.6**.
>
> **Q: What are EPD and JSD?**
> **A:** We will define these at first use in the revision.
>
> - **EPD:** Endpoint Distance — mean Euclidean distance between a lane's endpoint and its successor's start point.
> - **JSD:** Jensen–Shannon Divergence between generated and ground-truth histograms of a scalar attribute.
> - **Agent-JSD mean:** arithmetic mean of the six per-attribute agent JSDs (Figure 6); following ScenDream-L, Count JSD is excluded.
>
> **Q: What is generated during future outpainting?**
> **A:** Both **map** and **agents**. At each trigger we keep the observed region fixed and jointly sample unseen **frontier lane tokens** and **frontier agent tokens** for the next $64\ \mathrm{m} \times 64\ \mathrm{m}$ tile. New lanes and agents can thus appear. Newly generated agents are instantiated as interaction states (static state + motion code) and then simulated by $\Delta$Sim.
>
> **Q: Eq. (10): what is C, how is it computed, and why centering?**
> **A:** $C(s_t) \in \mathbb{R}^{K_{\mathrm{kd}}}$ is the vector of **one-step kinematic costs** over the discrete action vocabulary. For token $k$ with anchor $\Delta\xi_k = (\Delta x_k,\Delta y_k,\Delta\theta_k)$:
>
> $$C_k(s_t) = \sum_m w_m \, \phi_m(s_t, \Delta\xi_k),$$
>
> where soft penalties $\phi_m$ measure yaw rate, curvature, lateral/longitudinal acceleration, reverse motion, etc. $C_k$ is **low for feasible** and **high for implausible** actions.
>
> The centering operator:
>
> $\mathrm{ctr}(x)\_k = x\_k - \frac{1}{K\_{\mathrm{kd}}} \sum\_{j=1}^{K\_{\mathrm{kd}}} x\_j$
>
> is applied to both logits $\ell$ and costs $C(s_t)$. Centering is needed because the policy should learn the **relative ranking** of actions, not an arbitrary offset: a constant shift of all logits does not change the softmax, so the mean is unidentifiable; likewise for costs. Centering removes this nuisance and aligns **higher logit ↔ lower relative cost**, stabilizing DKAL and ensuring consistency with inference-time shaping $\tilde{\ell}_k = \ell_k - \lambda C_k(s_t)$.

---

> > ### Author Rebuttal · Reviewer_qTsg · 2026-04-03
> >
> > I thank the authors for their rebuttal. It was helpful and useful in providing clarifications and motivation on the method's design and individual components. Explanations were clear.
> >
> > I consider my previous questions sufficiently resolved and have no immediate new ones. Overall, it's a good paper. I think it will be nice to be presented to the community.

---

> > > ### Author Response · Authors · 2026-04-03
> > >
> > > We appreciate your careful review of our rebuttal and your positive acknowledgment. Your comments are very helpful, and we will carefully incorporate them into the final version.

---

### Official Review · Reviewer_p9W2 · 2026-03-10

**Soundness:** 3
**Presentation:** 3
**Significance:** 3
**Originality:** 3
**Overall Recommendation:** 4
**Confidence:** 1

**Summary:**

The paper presents a driving simulator called VectorWorld. Instead of predicting camera images, it predicts a compact map of the local driving scene: lane centerlines, lane connections, and nearby agents such as cars, pedestrians, and cyclists, all in a vector-graph form. The simulator works in a closed loop. It starts from one ego-centered 64m × 64m scene tile, and as the ego moves, it keeps the known part of the scene fixed and generates the new unseen area ahead. The goal is to make long driving rollouts possible without the simulator becoming too slow or unstable.

The main idea is to solve three practical problems in learned driving simulation: bad initialization at the start, slow diffusion sampling, and instability over long rollouts. To do this, the paper combines three pieces. First, it uses a motion-aware VAE that gives each agent not just a static state, but also a compact motion-history code, so newly inserted agents can be “warm-started” in a way that matches the behavior model. Second, it uses an edge-gated relational DiT to generate missing parts of the lane-agent graph while respecting lane topology and map-agent relations. Third, it uses a separate NPC behavior model called $\Delta$Sim to move the non-ego agents in a physically more stable way over time.

The paper’s main contributions are:
- A vector-graph world model for driving that generates and outpaints local lane-agent tiles during rollout, instead of relying on a static initial scene.
- A motion-aware interaction-state representation, where agents include motion-history information so inserted agents behave more smoothly at the start.
- An edge-gated relational DiT for graph generation, designed to preserve lane connectivity and relations between lanes and agents.
- A one-step, solver-free generation method based on interval-conditioned MeanFlow and JVP supervision, which is meant to make streaming generation fast enough for online use. The paper reports about 6 ms per 64m × 64m tile.
- A separate NPC behavior model, $\Delta$Sim, that improves long-horizon closed-loop stability by using physics-aligned actions.

Empirically, the paper reports better map-structure fidelity and better initialization validity than prior baselines on Waymo Open Motion and nuPlan, while also supporting 1 km+ closed-loop rollouts. It also shows that training a planner inside this simulator can improve success on hard stress-test settings.

**Compliance With Llm Reviewing Policy:**

Affirmed.

**Final Justification:**

Technically solid contribution with a well-motivated system design and good empirical validation. The rebuttal clarifies the role of each component and addresses my main concerns.

**Key Questions For Authors:**

no

**Limitations:**

yes

**Strengths And Weaknesses:**

$\textbf{Strengths}$:
The empirical evidence is a clear strength. The paper evaluates the methood in three modes that align with the deployment claim: (i) history-free initialization, (ii) online outpainting sampling latency and (iii) long-horizon feasibility or stability. On nuPlan and Waymo, VectorWorld improves several initialization metrics over prior vectorized scene generators, including endpoint distance and collision rate, and the latency plots show a one-step regime within the stated real-time budget. For what concerns the presentation the main claims, contributions, and evaluation setup are easy to locate.
None of the ingredients is entirely unprecedented in isolation: VAEs, DiT-style backbones, diffusion/flow, classifier-free guidance all have antecedents, but the combination is creative. Since I am not an expert on this particular field I cannot assess the field-specific novelty of the proposed approach.

$\textbf{Weaknesses}$: Because the system has many pieces, it is also hard to tell which part is responsible for which improvement. Some gains may come from the full pipeline rather than from one clearly new modeling idea. The simulator uses a compact vector representation of the world, not a full visual or very detailed map representation. The authors note that this leaves out some details such as road boundaries, curbs, and some traffic-control information and that this could matter in complex urban scenes.

---

> ### Author Rebuttal · Authors · 2026-03-26
>
> Thank you for the supportive review. We agree that the paper should make two things clearer: (i) which component addresses which specific problem, and (ii) the scope and rationale of the vector representation.
>
> **Q1: Clarify the purpose and gain of each key component.**
>
> **A:** Thank you. The main contribution of this work is a **real-time vectorized closed-loop simulator**. Each major module targets a distinct stage of the simulation loop, so their contributions can be evaluated independently:
>
> | Component | Quantitative gain | What it solves |
> |---|---|---|
> | Interaction-state VAE | Init. collision 9.45 → 4.69; jerk 16.6 → 9.6 | Provides motion-consistent initial states at $k=0$, removing cold-start artifacts |
> | Relation-aware EdgeDiT | Route validity 87.36 → 98.10; EPD 0.301 → 0.094 | Maintains lane topology and agent–lane relationships during iterative map extension |
> | JVP + two-time MeanFlow | At the same 5.6 ms budget, 1-step Flow validity 9.3% → 87.5%; EPD 8.52 → 0.269 | Corrects transport errors in large-step flow, enabling reliable single-step generation at real-time speed |
> | $\Delta$Sim | ADE 2.80 → 1.72; lateral-acc. violation 14.2% → 2.1% | Prevents physically implausible NPC behaviors from accumulating over long rollouts |
>
> In the revision, we will present these per-component results more prominently, showing the quantitative improvement and the specific problem addressed side by side.
>
> **Q2: Role and scope of the vector representation.**
>
> **A:** Thank you. We view the vector representation as a **deliberate design choice** that enables real-time performance, not a limitation. VectorWorld serves as a simulation backend for downstream planners. For this purpose, **lane centerlines, connectivity, and dynamic agent states** already capture the essential interaction structure needed for planning. This compact representation is precisely what makes **real-time streaming** possible. It is also practically effective: training PPO in VectorWorld improves the success rate from **25.7% to 56.0%** on the most challenging 1 km+ setting.
>
> We appreciate the observation that richer traffic-control information (e.g., curbs, detailed signal phases) can matter in dense urban scenes. Our current representation already includes **traffic-light states when available**, and the interface can be extended with additional attributes without changing the overall architecture. Importantly, much of the difficulty in autonomous driving arises from **multi-agent interaction**, which is primarily governed by lane geometry, connectivity, and the motion of nearby agents. The resulting trajectories can then be mapped back to an HD map through standard lane-matching procedures. This planner-oriented abstraction is also adopted by recent vectorized simulation systems such as **SLEDGE [1], VADv2 [2], and ScenDream-L [3]**, and it is a key reason our system achieves real-time performance.
>
> [1] Chitta, K., Dauner, D., & Geiger, A. (2024, September). Sledge: Synthesizing driving environments with generative models and rule-based traffic. In European Conference on Computer Vision (pp. 57-74). Cham: Springer Nature Switzerland.
>
> [2] Jiang, B., Chen, S., Gao, H., Liao, B., Zhang, Q., Liu, W., & Wang, X. (2026). VADv2: End-to-End Autonomous Driving via Probabilistic Planning. In The Fourteenth International Conference on Learning Representations.
>
> [3] Rowe, L., Girgis, R., Gosselin, A., Paull, L., Pal, C., & Heide, F. (2025). Scenario Dreamer: Vectorized latent diffusion for generating driving simulation environments. In Proceedings of the IEEE/CVF Conference on Computer Vision and Pattern Recognition (pp. 17207-17218).

---

> > ### Author Rebuttal · Reviewer_p9W2 · 2026-04-01
> >
> > Thank you for the clear and constructive rebuttal. I appreciate the effort to better assign the roles of each component; this addressed my main concerns.
> >
> > I am satisfied with the responses and will keep my original score. That said, as this topic is somewhat outside my area of expertise, I prefer to defer to the judgment of reviewers with more domain knowledge than me for the final decision.

---

> > > ### Author Response · Authors · 2026-04-02
> > >
> > > Thank you very much for your thoughtful acknowledgment and kind recognition of our rebuttal.
> > > Your comments are highly valuable to us, and we will carefully take them into full consideration in the final version.

---

### Official Review · Reviewer_Gpcp · 2026-03-13

**Soundness:** 3
**Presentation:** 3
**Significance:** 3
**Originality:** 3
**Overall Recommendation:** 5
**Confidence:** 4

**Summary:**

This paper proposes VectorWorld, a vectorized generative simulation framework for autonomous driving. VectorWorld comprises two methodological components: an efficient streaming latent diffusion model for agent+map generation (as 64x64m “tiles”), and a return-conditioned behaviour model for agent simulation. VectorWorld can be seen as a direct extension of Scenario Dreamer (Rowe et al. CVPR 2025), with novel design choices that improve the streaming efficiency and long-horizon closed-loop stability. Specifically, the authors propose (1) real-time sampling via a JVP-augmented MeanFlow objective, (2) augmenting the initial agent states with motion histories to improve compatibility with the downstream behaviour model, and (3) several design choices (Film conditioning, hybrid discrete/continuous action space, and kinematic feasibility loss) to improve closed-loop stability of the agent behaviour model over long-horizons.

**Compliance With Llm Reviewing Policy:**

Affirmed.

**Final Justification:**

The rebuttal addressed my concerns and I believe the method is sufficiently good, novel and interesting to the community.

**Key Questions For Authors:**

Questions:
- Why is the RTG column not checkmarked for ctrl-sim? ctrl-sim is return-conditioned.
- Was there a good reason to ignore the Urban Planning metrics in the main paper? If not, can you add those results to the main paper, and factor those into the overall discussion? I would also prefer that Figure 6 uses either Frechet distance or an Urban Planning metric (eg, SLEDGE used Frechet Reach in their scaling experiments), as that is a better metric that characterizes lane graph fidelity.
- Do you sample multiple times at each outpainting step and choose one? If so, that should be clearly stated. If not, that should also be clearly stated as that would be an improvement over prior approaches in efficiency (SLEDGE and Scenario Dreamer both sample 8 inpainting “candidates” at each step and choose 1).

**Limitations:**

Yes the authors did.

**Strengths And Weaknesses:**

Strengths
- The problem is well-motivated and timely. Closed-loop vectorized generative simulation for autonomous driving is an area of growing interest, and achieving efficient, stable long-horizon rollouts remains an open challenge.
- The paper is overall well-written and the presentation of the approach is clear.
- The empirical results on the Waymo and nuPlan datasets demonstrate improvements over prior work and the comprehensive planning results demonstrate the practical utility of the proposed approach.
- The finding that training within VectorWorld yields improved PPO policy performance is a novel and interesting result. To my knowledge, prior works (SLEDGE, DriveSceneGen, Scenario Dreamer) have not shown this.

Weaknesses
- My primary concern is the choice of evaluation metrics for lane graph fidelity. Route length and endpoint distance are, in my view, among the least informative metrics for this purpose. Fréchet distance and the Urban Planning metrics are better suited to evaluating whether generated scenes faithfully reflect the real-world distribution. Endpoint distance, in particular, can be gamed without the lane graph actually capturing the target distribution. The Urban Planning results are currently reported in the Appendix, where Scenario Dreamer appears competitive with VectorWorld. These results should be moved to the main paper and incorporated into the overall discussion. Figure 6 would also be more convincing with Fréchet distance or an Urban Planning metric on the y-axis. SLEDGE, for instance, used Fréchet Reach for their scaling experiments.
- This paper proposes a lot of additional complexity over Scenario Dreamer. Rather than one self-contained contribution, this paper has a lot of minor contributions that taken together, yield a system that outperforms prior work.
- Following on the previous point, many of the ablations supporting key design decisions are deferred to the Appendix. The paper would be improved by moving the most important ablations into the main paper and moving less important design choices (e.g., inference-time kinematic shaping, edge-conditioned bias) to the Appendix. Currently, the empirical justification for the core methodological claims is not apparent from reading the main paper.
- The title does not reflect the autonomous driving focus of the work. Several design decisions, eg, kinematic feasibility, are specific to the driving domain, and a general-sounding title risks misleading readers into expecting a more broadly applicable world model.

---

> ### Author Rebuttal · Authors · 2026-03-26
>
> Thank you for your constructive review. We will revise the manuscript accordingly. Our point-by-point responses follow:
>
> **Q: Lane-fidelity metrics / Urban-Planning metrics**
>
> **A:** We will move the Urban-Planning topology metrics into the main paper and discuss them with FD.
>
> A direct comparison with ScenDream-L is below (lower is better):
>
> | Metric | nuPlan ScenDream-L | nuPlan Ours | Waymo ScenDream-L | Waymo Ours |
> |---|---:|---:|---:|---:|
> | FD | **0.67** | 0.98 | 1.38 | **0.94** |
> | Conn. | 0.18 | **0.09** | **0.03** | 0.23 |
> | Dens. | 0.43 | **0.41** | 0.95 | **0.57** |
> | Reach | **0.03** | 0.09 | **0.28** | 0.37 |
> | Conve. | **0.33** | 0.79 | **2.05** | 2.63 |
> | Count JSD | 0.011 | **0.004** | 0.029 | **0.001** |
> | EPD | 0.250 | **0.078** | 0.210 | **0.094** |
> | Init. Coll. | 9.30 | **3.01** | 4.80 | **4.69** |
>
> The core contribution of VectorWorld is a **real-time vectorized closed-loop simulation system**. Accordingly, the most deployment-critical properties are local lane continuity, density calibration, and initialization validity. On these, VectorWorld is consistently strong: it improves Count JSD, EPD, and initialization collisions on both datasets; it also improves FD on Waymo, and Conn./Dens. on nuPlan.
>
> ScenDream-L remains better on a subset of topology metrics (FD on nuPlan; Reach/Conve. and Waymo Conn.), but the remaining gaps are small in absolute terms. The Reach gaps are $0.06$ on nuPlan and $0.09$ on Waymo; after removing the standard $\times 10$ scaling, the raw Conve. gaps are only $0.046$ and $0.058$, and the raw Waymo Conn. gap is $0.020$. In practice, this has little negative effect on simulator use, because closed-loop performance depends primarily on **local successor continuity and agent-map consistency**, while the remaining gap is concentrated on **minor branches / connectors in the compact keypoint-graph evaluation**, rather than on a general loss of realism. We will make this trade-off explicit.
>
>
> **Q: Figure 6 metric**
>
> **A:** Thank you. We will revise Figure 6 to use FD and a topology metric (e.g., Fréchet Reach / Urban-Planning metric) on the quality axis, so the scaling plot directly reflects lane-graph fidelity under the latency budget:
>
> | Method | FD ↓ | Fréchet Reach ↓ | Agent JSD Mean ↓ | Time (ms) ↓|
> | :---: | :---: | :---: | :---: | :---: |
> | ScenDream-L | 1.38 | **0.28** | 0.205 | 160 |
> | Ours (1 step) | 1.71 | 0.63 | 0.294 | **5.6** |
> | Ours (3 steps) | 1.56 | 0.51 | 0.251 | 12.8 |
> | Ours (5 steps) | 1.42 | 0.48 | 0.212 | 21.3 |
> | **Ours (Flow)** | **0.94** | 0.37 | **0.098** | 70 |
>
> *Note: Following Scenario Dreamer, count is excluded from Agent JSD Mean averaging.*
>
>
> **Q: What is the central novelty relative to prior vectorized simulators, especially ScenDream-L?**
>
> **A:** Thank you. The paper-level novelty is not a collection of isolated tweaks. **VectorWorld is the first real-time vectorized closed-loop simulator** that jointly provides **history-aligned initialization, solver-free frontier outpainting, and kilometer-scale stable rollouts**. This unified operating point is realized by three tightly coupled contributions:
>
> | Closed-loop requirement | VectorWorld contribution | Representative evidence |
> |---|---|---|
> | History-compatible initialization | Interaction-state VAE | Init. coll. $9.45 \rightarrow 4.69$; jerk $16.6 \rightarrow 9.6$ |
> | Real-time frontier outpainting | Relation-aware EdgeDiT + JVP MeanFlow | $5.6$ ms/tile; 1-step Flow valid $9.3\% \rightarrow 87.5\%$; EPD $8.52 \rightarrow 0.269$ |
> | Long-horizon NPC stability | 1-pass physics-aligned $\Delta$Sim | ADE $2.80 \rightarrow 1.72$; lat.-acc. violation $14.2\% \rightarrow 2.1\%$; forward passes $2 \rightarrow 1$ |
>
> ScenDream-L is the closest prior vectorized generator, but it does not realize this combined operating point. We will revise the introduction and contribution list to foreground this **single system contribution**, with the lower-level mechanisms presented as internal designs of these three modules.
>
> **Q: Key ablations are in the appendix**
>
> **A:** Agreed. We will move the key ablations to the main paper: warm-start, relation-aware generation, and one-step MeanFlow + $\Delta$Sim (Tables 9, 10, 11, 13).
>
> **Q: Title scope**
>
> **A:** We will make the AD scope explicit, e.g., **"VectorWorld: Efficient Streaming World Model for Autonomous-Driving Simulation via Diffusion Flow on Vector Graphs."**
>
> **Q: RTG column in Table 3**
>
> **A:** Thank you. This was a labeling mistake. Ctrl-Sim is return-conditioned. Our intended distinction was **1-pass RTG** vs generic RTG conditioning. We will rename/split the column and mark Ctrl-Sim correctly.
>
> **Q: Multiple samples per outpainting step?**
>
> **A:** Yes. For closed-loop robustness, we use 8 candidates and choose the best valid one. Latency is 5.6–6 ms per candidate (one MeanFlow update + decode). For an 8-candidate batch: **Ours 44.8 ms vs ScenDream-L 1280 ms** (~28× faster). We will report both.

---

> > ### Author Rebuttal · Reviewer_Gpcp · 2026-04-03
> >
> > I am happy with the rebuttal of the authors which addressed my concerns. I will be raising my score.

---

> > > ### Author Response · Authors · 2026-04-03
> > >
> > > Thank you for your careful reading of our rebuttal and for your positive assessment. We greatly appreciate your feedback and will incorporate it carefully in the final version.

---

### Decision · Program_Chairs · 2026-04-30

**Decision:**

Accept (spotlight)

**Comment:**

This paper proposes a vectorized generative simulator for autonomous driving, proposing a streaming latent diffusion model and behavioral model for agent simulation. Reviewers appreciated the writing, motivation, method, and results, especially policy training demonstration utilizing VectorWorld. Some concerns were raised regarding the metric choices, ablations (which are in appendix, making the paper not self-contained), and strong ablations demonstration which components were most crucial. Post-rebuttal, all reviewers recommended accepting this paper and mentioned any concerns have been addressed, including with score increases. Therefore, I recommend acceptance of this paper and encourage the authors to revise based on the reviews/rebuttals for the camera-ready version.